# Understanding meteorological influences on PM$_{2.5}$ concentrations across China: a temporal and spatial perspective

**Ziyue Chen[1,2], Xiaoming Xie[1], Jun Cai[3], Danlu Chen[1], Bingbo Gao[4], Bin He[1,2],**

**Nianliang Cheng[5], Bing Xu[3]***

[1] College of Global Change and Earth System Science, Beijing Normal University, 19 Xinjiekouwai Street, Haidian, Beijing, 100875, China

[2] Joint Center for Global Change Studies, Beijing 100875, China

[3] Ministry of Education Key Laboratory for Earth System Modeling, Department of Earth System Science, Tsinghua University, Beijing 100084, China

[4] National Engineering Research Center for Information Technology in Agriculture, 11 Shuguang Huayuan Middle Road, Beijing 100097, China

[5] Beijing Municipal Environmental Monitoring Center, Beijing 100048, China

## Abstract

With frequent air pollution episodes in China, growing research emphasis has been put on quantifying meteorological influences on PM$_{2.5}$ concentrations. However, these studies mainly focus on isolated cities whilst meteorological influences on PM$_{2.5}$ concentrations at the national scale have yet been examined comprehensively. This research employs the CCM (Cross Convergent Mapping) method to understand the influence of individual meteorological factors on local PM$_{2.5}$ concentrations in 188 monitoring cities across China. Results indicate that meteorological influences on PM$_{2.5}$ concentrations are of notable seasonal and regional variations. For the heavily polluted North China region, when PM$_{2.5}$ concentrations are high, meteorological influences on PM$_{2.5}$ concentrations are strong. The dominant meteorological influence for PM$_{2.5}$ concentrations varies across locations and demonstrates regional similarities. For the most polluted winter, the dominant meteorological driver for local PM$_{2.5}$ concentrations is mainly the wind within the North China region whilst precipitation is the dominant meteorological influence for most coastal regions. At the national scale, the influence of temperature, humidity and wind on PM$_{2.5}$ concentrations is much larger than that of other meteorological factors. Amongst eight factors, temperature exerts the strongest and most stable influence on national PM$_{2.5}$ concentrations in all seasons. Due to notable temporal and spatial differences in meteorological influences on local PM$_{2.5}$ concentrations, this research suggests pertinent environmental projects for air quality improvement should be designed accordingly for specific regions.

**Keywords: PM$_{2.5}$; Meteorological factors; Causality analysis; CCM**

---

* Corresponding author.    Email address: bingxu@tsinghua.edu.cn.    Telephone No. 0086(10) 62773906

## Introduction

With rapid social and economic growth in China, both the government and residents are placing more and more emphasis on the sustainability of the ambient environment, and air quality has become one of the most concerned social and ecological issues. Recently, the frequency of air pollution episodes with high $PM_{2.5}$ concentrations and the number of cities influenced by $PM_{2.5}$ pollution have increased notably in China since 2013. Statistical records from the national air quality publishing platform (http://113.108.142.147:20035/emcpublish/) revealed that $PM_{2.5}$ induced pollution events occurred in 25 provinces and more than 100 middle-large cities whilst there were on average 30 days with hazardous $PM_{2.5}$ concentrations for each monitoring city in 2014.

High $PM_{2.5}$ concentrations not only influence people's daily life (e.g. high $PM_{2.5}$ concentrations caused severe traffic jam), but also severely threaten the health of residents that suffer from polluted air quality. Recent studies have suggested that airborne pollutants, $PM_{2.5}$ in particular, are closely related to cardiovascular disease-related mortality (Garrett and Casimiro, 2011, Li et al., 2015a), emergency room visits (Qiao et al., 2014), all year non-accidental mortality (Pasca et al., 2014) and cardiovascular mortality (Lanzinger et al., 2015). Due to its strong negative influences on public health , scholars have been working towards a better understanding of sources (Guo et al., 2012; Zhang et al., 2013;Gu et al., 2014; Liu et al., 2014; Cao et al., 2014), characteristics (Wei et al., 2012; Zhang et al., 2013; Hu et al., 2015; Zhang, F. et al., 2015; Zhen et al., 2016; Zhang et al., 2016) and seasonal variations (Cao et al., 2012; Shen et al., 2014; Yang and Christakos, 2015; Wang et al., 2015; Chen et al., 2015; Chen, Y. et al. 2016; Chen, Z. et al., 2016) of $PM_{2.5}$ and other airborne pollutants. Meanwhile, large-scale research on the variation and distribution of $PM_{2.5}$ has been conducted using a variety of remote sensing sources and spatial data analysis methods (Ma et al., 2014; Kong et al., 2016).

One key issue for air quality research is to find the source and influencing factors for airborne pollutants. Although quantitative contributions of different sources (e.g. coal burning and automobile exhaust) to airborne pollutants remain controversial, meteorological influences on airborne pollutants have been examined in depth by more and more scholars. Recent studies conducted in different countries indicated that $PM_{2.5}$ were closely related to temperature (Pearce et al., 2011; Yadav et al., 2014; Grundstrom

et al., 2015),wind speed (Galindo et al., 2011; El-Metwally and Alfaro, 2013; Yadav et al., 2014) and precipitation (Yadav et al., 2014). Meanwhile, meteorological influences on $PM_{2.5}$ concentrations across China have also become a hot research topic. Yao (2017) revealed a generally negative correlation between evaporation and $PM_{2.5}$ concentrations in a series of cities within the North China plain. Huang et al. (2015) and Yin et al., (2016) found a negative influence of sunshine duration and a positive influence of relative humidity on $PM_{2.5}$ concentrations in Beijing. Li et al. (2015) suggested that air pressure and temperature was positively correlated with $PM_{2.5}$ concentrations in Chengdu. For Nanjing (Chen, T. et al., 2016) and Hong Kong (Fung et al., 2014), precipitation exerted a strong influence on $PM_{2.5}$ concentrations in winter, when the influence of wind speed on $PM_{2.5}$ concentrations was weak. Meanwhile, wind speed exerted a major influence on $PM_{2.5}$ concentrations in Beijing in winter. Through experiments, Guo (et al., 2016) found that the influence of precipitation on $PM_{2.5}$ concentrations in Xi'an was weaker than that in Guangzhou. Zhang et al. (2015b) quantified the correlations between meteorological factors and main airborne pollutants in three megacities, Beijing, Shanghai and Guangzhou, and pointed out that the influences of meteorological factors on the formation and concentrations of airborne pollutants varied significantly across seasons and geographical locations. Chen, Z. et al. (2017) quantified the meteorological influences on local $PM_{2.5}$ concentrations in the Beijing-Tianjin-Hebei region and revealed that wind, humidity and solar radiation were major meteorological factors that significantly influenced local $PM_{2.5}$ concentrations in winter. These studies revealed the correlations between $PM_{2.5}$ concentrations and a diversity of meteorological factors in some specific cities. However, findings from these studies conducted at a local scale cannot reveal regional and national patterns of meteorological influences on $PM_{2.5}$ concentrations in China. In addition, these studies mainly employed short-term observation data (e.g. one season or one year) and thus revealed characteristics of meteorological influences on $PM_{2.5}$ concentrations may be biased by inter-annual variations.

Due to the diversity of meteorological factors and complicated interactions between them, Pearce et al (2011) suggested that multiple models and methods should be comprehensively employed to quantify the influence of meteorological factors on local airborne pollutants. Due to complicated interactions between different factors, Sugihara

et al. (2012) suggested that correlation analysis between two variables in a complicated ecosystem might lead to mirage correlations and the extracted correlation coefficient between two variables could be influenced significantly by other variables in the ecosystem. To better examine the coupling between two variables in a complicated system, Sugihara et al. (2012) proposed a CCM (Cross Convergent Mapping) method to qualify the bi-direction coupling between two variables without the influence from other variables. Therefore, the CCM method can effectively remove mirage correlations and extract reliable causality between two variables. Our previous research (Chen, Z., 2017) found that the CCM (Cross Convergent Mapping) method performed better in quantifying the influence of individual meteorological factors on $PM_{2.5}$ concentrations than traditional correlation analysis through comprehensive comparison. However, this study mainly focused on the meteorological influences on $PM_{2.5}$ concentrations in a specific region. As pointed out by some scholars, interactions between meteorological factors and airborne pollutants are of great variations for different regions, yet most relevant studies have been conducted at the local or regional scale. China is a large country, including many regions with completely different air pollution levels, geographical conditions and meteorological types. To better understand the variations of meteorological influences on $PM_{2.5}$ concentrations, a comparative study at the national scale is required.

According to these challenges, this research aims to analyze and compare the influence of individual meteorological factors on $PM_{2.5}$ concentrations across China. Based on the CCM causality analysis, we quantified the influence of eight meteorological factors on $PM_{2.5}$ concentrations in 188 monitoring cities across China using the observation data from March, 2014 to February, 2017. To comprehensively understand the spatio-temporal patterns of meteorological influences on $PM_{2.5}$ concentrations across China, we a). investigated comprehensive meteorological influences on $PM_{2.5}$ concentrations for 37 regional representative cities, b) extracted the seasonal dominant meteorological factor for each monitoring city, and c) conducted a comparative statistics of the influence of different meteorological factors on $PM_{2.5}$ concentrations at the national scale.

**2 Materials**
**2.1 Data sources**
**2.1.1 PM$_{2.5}$ data**
PM$_{2.5}$ data are acquired from the website PM25.in. This website collects official data of
PM$_{2.5}$ concentrations provided by China National Environmental Monitoring Center
(CNEMC) and publishes hourly air quality information for all monitoring cities. Before
Jan 1st, 2015, PM25.in publishes data of 190 monitoring cities. Since Jan 1$^{st}$, 2015, the
number of monitoring cities has increased to 367. By calling specific API (Application
Programming Interface) provided by PM25.in, we collect hourly PM$_{2.5}$ data for target
cities. The daily PM$_{2.5}$ concentrations for each city is calculated using the averaged value
of hourly PM$_{2.5}$ concentrations measured at all available local observation stations. For a
consecutive division of different seasons and multiple-year analysis, We collected PM$_{2.5}$
data from March 1$^{st}$, 2014 to February 28$^{th}$, 2017 for the following analysis.
**2.1.2 Meteorological data**
The meteorological data for these monitoring cities are obtained from the "China
Meteorological Data Sharing Service System", part of National Science and Technology
Infrastructure. The meteorological data are collected through thousands of observation
stations across China. Previous studies (Zhang et al., 2015b; Pearce et al., 2011; Yadav et
al., 2014) revealed that such meteorological factors as relative humidity, temperature,
wind speed, wind direction, solar radiation, evaporation, precipitation, and air pressure
may be related to PM$_{2.5}$ concentrations. Therefore, to comprehensively understand
meteorological driving forces for PM$_{2.5}$ concentrations in China, all these potential
meteorological factors were selected as candidate factors. To better quantify the role of
these meteorological factors in affecting local PM$_{2.5}$ concentrations, these factors are
further categorized into some sub-factors: evaporation (small evaporation and large
evaporation), temperature (daily max temperature, mean temperature, minimum
temperature, and largest temperature difference for the day), precipitation (total
precipitation from 8am-8pm, total precipitation from 8pm-8am and total precipitation for
the day), air pressure (daily max pressure, mean pressure and minimum pressure),
humidity (daily mean and minimum relative humidity), radiation (sunshine duration for
the day, short for SSD), wind speed (mean wind speed, max wind speed andextreme
wind speed), wind direction (max wind direction for the day). Some meteorological
factors are briefly explained here. Evaporation indicates the amount of
evaporation-induced water loss during a certain period and is usually calculated using the
depth of evaporated water in a container. For this research, small (large) evaporation
indicates the amount of evaporated water measured using a container with a diameter of
10cm (30cm) during 24 hours (unit: mm). Generally, the measured values using the two
types of equipment are of slight differences. SSD represents the hours of sunshine
measured during a day for a specific location on earth. The max wind speed indicates
the max mean wind speed during any 10 minutes within a day's time. The extreme
wind speed indicates the max instant (for 1s) wind speed within a day's time. The
max wind direction indicates the dominant wind direction for the period with the
max wind speed. As there are one or more observation stations for each city, the daily
value for each meteorological factor for each city was calculated using the mean value of
all available observation stations within the target city. To conduct time series
comparison, we also collected meteorological data from March 1st, 2014 to February 28th,

174   2017.

**2.2 Study sites**
For a comprehensive understanding of meteorological influences on local PM$_{2.5}$
concentrations across China,all monitoring cities (except for Liaocheng and Zhuji,
where continuous valid meteorological data were not available) during the study period
were selected for this research. The 188 cities included most major cities (Beijing,
Shanghai, Guangzhou, etc.) in China. For regions (e.g. Beijing-Tianjin-Hebei region)
with heavy air pollution, the density of monitored cities was much higher than that in
regions with good air quality.
**3 Methods**
Due to complicated interactions in the atmospheric environment, it is highly difficult to
quantify the causality of individual meteorological factors on PM$_{2.5}$ concentrations
through correlation analysis. Instead, a robust causality analysis method is required.
To extract the coupling between individual variables in complex systems, Sugihara et al.
(2012) proposed a convergent cross mapping (CCM) method. Different from Granger
causality (GC) analysis (Granger, 1980), the CCM method is sensitive to weak to
moderate coupling in ecological time series. By analyzing the temporal variations of two
time-series variables, their bidirectional coupling can be featured with a convergent map.
If the influence of one variable on the other variable is presented as a convergent curve
with increasing time series length, then the causality is detected; If the curve
demonstrates no convergent trend, then no causality exists. The predictive skill (defined
as $\rho$ value), which ranges from 0 to 1, suggests the quantitative causality of one
variable on the other.
The principle of CCM algorithms is briefly explained as follows (Luo et al. 2014). Two
time series $\{X\}= [X(1), \ldots, X(L)]$ and $\{Y\} = [Y(1), \ldots, Y(L)]$ are defined as the temporal
variations of two variables $X$ and $Y$. For r = S to L (S < L), two partial time series
$[X(1), \ldots, X(L_P)]$ and $[Y(1), \ldots, Y(L_P)]$ are extracted from the original time series (r is the
current position whilst S is the start position in the time series). Following this, the
shadow manifold $M_X$ is generated from $\{X\}$, which is a set of lagged-coordinate vectors
x(t) = <X(t), X(t-$\tau$), ..., X(t-(E-1)$\tau$)> for t = 1+(E-1) $\tau$ to t = r. To generate a
cross-mapped estimate of Y(t) ($\hat{Y}$(t)|$M_X$), the contemporaneous lagged-coordinate vector
on $M_X$, x(t) is located, and then its E+1 nearest neighbors are extracted, where E+1 is the
minimum number of points required for a bounding simplex in an E-dimensional space
(Sugihara and May, 1990). Next, the time index of the E+1 nearest neighbors of x(t) is
denoted as $t_1$, ..., $t_{E+1}$. These time index are used to identify neighbor points in $Y$ and then
estimate Y(t) according to a locally weighted mean of E+1 $Y(t_i)$ values (Equation 1).
$$\hat{Y}(t)|M_X = \sum_{i=1}^{E+1} w_i Y(t_i)$$
(E1)

Where $w_i$ is a weight calculated according to the distance between $X$(t) and its i$^{th}$ nearest
neighbor on $M_X$. $Y(t_i)$ are contemporaneous values of Y. The weight $w_i$ is determined according to
Equation 2.
$$w_i = u_i \bigg/ \sum_{j=1}^{E+1} u_j$$
(E2)

Where $u_i = e^{-d[x(t),x(t_i)]/d[x(t),x(t_1)]}$ whilst $d[x(t), x(t_i)]$ represents the Euclidean distance between
two vectors.
In our previous research, interactions between the air quality in neighboring cities (Chen,
Z. et al., 2016), and bidirectional coupling between individual meteorological factors and
$PM_{2.5}$ concentrations (Chen, Z. et al., 2017) were quantified effectively using the CCM
method. By comparing the performance of correlation analysis and CCM method, Chen,
Z. et al. (2017) suggested that correlation analysis may lead to a diversity of biases due
to complicated interactions between individual meteorological factors. Firstly, some
mirage correlations (two variables with a moderate correlation coefficient) extracted
using the correlation analysis were revealed effectively using the CCM method (the $\rho$
value between two variables was 0). Secondly, some weak coupling, which was hardly
detected using the correlation analysis (the correlation between the two variables were
not significant), was extracted using the CCM method (a small $\rho$ value). Meanwhile,
as Sugihara et al. (2012) suggested, the correlation between two variables could be
influenced significantly by other agent variables and thus the value of correlation
coefficient between two variables could not reflect the actual causality between them.
Chen et al. (2017) further revealed that the correlation coefficient between individual
meteorological factors and $PM_{2.5}$ concentrations was usually much larger than the $\rho$
value. This indicated that the causality of individual meteorological factors on $PM_{2.5}$
concentrations was generally overestimated using the correlation analysis, due to the
influences from other meteorological factors. In this case, the CCM method is an
appropriate tool for quantifying bidirectional interactions between $PM_{2.5}$ concentrations
and individual meteorological factors in complicated atmospheric environment.
**4 Results**
Seasonal variations of $PM_{2.5}$ concentrations have been revealed in Beijing (Chen et al.,
2015; Chen, Y. et al., 2016; Chen, Z. et al., 2016), Nanjing (Shen et al., 2014), Shandong
Province (Yang and Christakos, 2015) and the Beijing-Tianjin-Hebei region (Wang et al.
2015; Chen, Z. et al., 2017). In addition to these local and regional studies, Cao et al.
(2012) further compared seasonal variations of $PM_{2.5}$ concentrations in seven southern
cities (Chongqing, Guangzhou, Hong Kong, Hangzhou, Shanghai, Wuhan, and Xiamen)
and seven northern cities (Beijing, Changchun, Jinchang, Qingdao, Tianjin, Xi'an, and
Yulin) across China.   Hence, the research period was divided into four seasons.
According to traditional season division for China, spring was set as the period between
March 1st, 2014 and May 31st, 2014; summer was set as the period between June 1st,
2014 and August 31st, 2014; autumn was set as the period between September 1st, 2014

and November 30th, 2014; and winter was set as the period between December 1st, 2014 and February 28th, 2015. For each city, the bidirectional coupling between individual meteorological factors and PM$_{2.5}$ concentrations in different seasons was analyzed respectively using the CCM method. The CCM method is highly automatic and only few parameters need to be set for running this algorithm: E (number of dimensions for the attractor reconstruction), $\tau$ (time lag) and b (number of nearest neighbors to use for prediction). The value of E can be 2 or 3. A larger value of E produces more accurate convergent maps. The variable b is decided by E (b = E + 1). A small value of $\tau$ leads to a fine-resolution convergent map, yet requires much more processing time. Through experiments, we found that the final results were not sensitive to the selection of parameters and different parameters mainly exerted influences on the presentation effects of CCM. In this research, to acquire optimal interpretation effects of convergent cross maps, the value of $\tau$ was set as 2 days and the value of E was set 3. For each meteorological factor, its causality coupling with PM$_{2.5}$ concentrations can be represented using a convergent map. Since it is not feasible to present all these convergent maps here, we simply display some exemplary maps to demonstrate how CCM works (Fig 1). As a heavily polluted city, we presented the interactions between PM$_{2.5}$ concentrations and meteorological factors in Beijing in winter, when the local PM$_{2.5}$ concentration was the highest, as an example. Four major meteorological factors, wind, humidity, radiation and temperature, which exerted much stronger influences on PM$_{2.5}$ concentrations than other factors, were employed. Due to the strong bidirectional coupling between PM$_{2.5}$ concentrations and these meteorological factors, Figure 1 not only demonstrates how CCM output could be interpreted, but also provides readers with a general comparison of the magnitude of simultaneous influences of different meteorological factors on the local PM$_{2.5}$ concentration and its feedback effects.

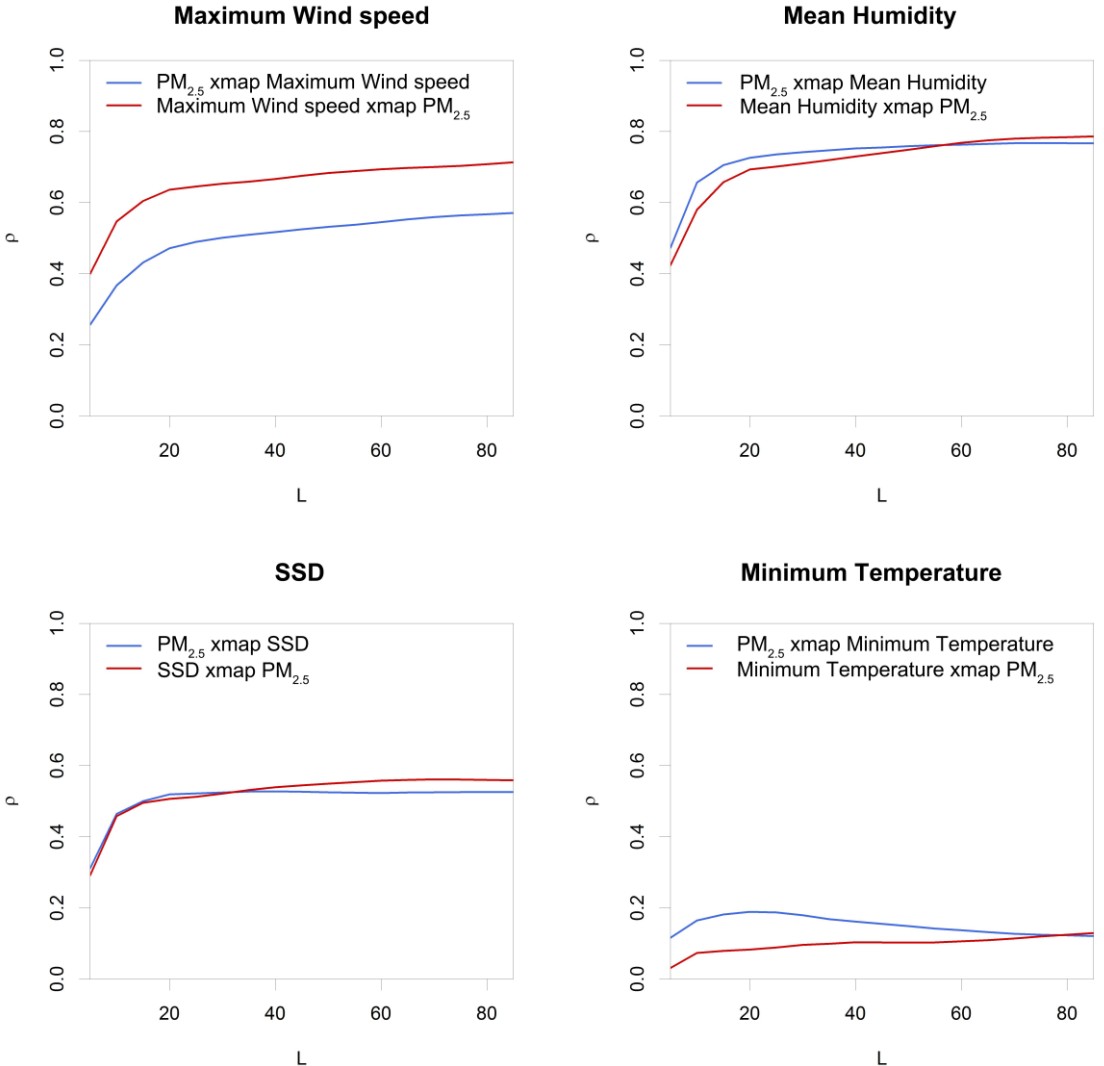

275

**Fig 1. Illustrative CCM results to demonstrate the bidirectional coupling between meteorological factors and PM$_{2.5}$ concentrations in Beijing (2014, winter)**

$\rho$ : **predictive skills.** $L$ : **the length of time series. A xmap B stands for convergent cross mapping B from A, in other words, the causality of variable B on A. For instance, PM$_{2.5}$ xmap meanRHU stands for the causality of meanRHU on PM$_{2.5}$ concentrations. meanRHU xmap PM$_{2.5}$ stands for the feedback effect of PM$_{2.5}$ on meanRHU concentrations.** $\rho$ **indicates the predictive skills of using meanRHU to retrieve PM$_{2.5}$ concentrations.**

According to Fig 1, one can see that the quantitative influence of individual meteorological factors on PM$_{2.5}$ was well extracted using the CCM method whilst the feedback effect of PM$_{2.5}$ on specific meteorological factors was revealed as well. For Beijing, meanRHU and maxWIN exerted a strong influence on local PM$_{2.5}$ concentrations in Winter ( $\rho$ > 0.4) whilst SSD and minTEM also had a weaker influence on local PM$_{2.5}$ concentrations. ( $\rho$ close to 0.2 ). On the other hand, high PM$_{2.5}$

concentrations had an even stronger feedback influence on meanRHU, maxWIN and
SSD ($\rho$ close to 0.6) whilst $PM_{2.5}$ had little influence on minTEM ($\rho$ close to 0). The
bidirectional coupling between $PM_{2.5}$ concentrations and individual meteorological
factors provides useful reference for a better understanding of the form and development
of $PM_{2.5}$-induced air pollution episodes. For Beijing, low wind speed (high humidity and
low SSD) in winter results in high $PM_{2.5}$ concentrations, which in turn causes lower wind
speed (higher humidity and lower SSD). In consequence, $PM_{2.5}$ concentrations are
increased further by the changing wind (humidity and SSD) situation. This mechanism
causes a quickly rising $PM_{2.5}$ concentrations, which brings the atmospheric environment
to a comparatively stable status. In this case, persistent high-concentration $PM_{2.5}$ is
unlikely to disperse and usually lasts for a long period in this region. Similarly,
bidirectional interactions between $PM_{2.5}$ concentrations and other meteorological factors
can as well be quantified using the CCM method. Since the main aim of this research is
to understand the influence of individual meteorological factors on $PM_{2.5}$ concentrations
across China, the feedback effect of $PM_{2.5}$ concentrations on specific meteorological
factors is not explained in details herein.
The $\rho$ value is a direct indicator of quantitative causality. For this research, the
maximum $\rho$ value of all sub-factors in the same category was used as the causality
of this specific meteorological factor on $PM_{2.5}$ concentrations. E.g. for a specific city, the
maximum $\rho$ value of max temperature, mean temperature, minimum temperature, and
largest temperature difference for the day is used as the influence of temperature on local
$PM_{2.5}$ concentrations. For this research, we collected meteorological and $PM_{2.5}$ data for
three consecutive years. To avoid the analysis of inconsecutive time series, which may
influence the CCM result, we did not calculate the general influence of individual
meteorological factors on $PM_{2.5}$ concentrations during 2014-2016 by analyzing three
isolated periods (e.g. April- June, 2014, April-June, 2015, and April- June, 2016) as a
complete data set. Instead, for each city, we quantified the influence of individual
meteorological factors on $PM_{2.5}$ concentrations for each season in 2014, 2015 and 2016
respectively and calculated the mean $\rho$ value during 2014-2016 for each city.
**4.1 Comprehensive meteorological influences on PM$_{2.5}$ concentrations in some**
**regional representative cities**
When the $\rho$ value for each meteorological factor was calculated, a wind rose, which
presents the quantitative influences of all individual meteorological factors on PM$_{2.5}$
concentrations, can be produced for each city. It is not feasible to present all 188 wind
roses simultaneously, due to severe overlapping effects. Thus, considering the
social-economic factors, 37 regional representative cities (including all 31 provincial
capital cities in mainland China), which are the largest and most important cities for
specific regions, were selected to produce a wind rose map of meteorological influences
on PM$_{2.5}$ concentrations across China (Fig 2).

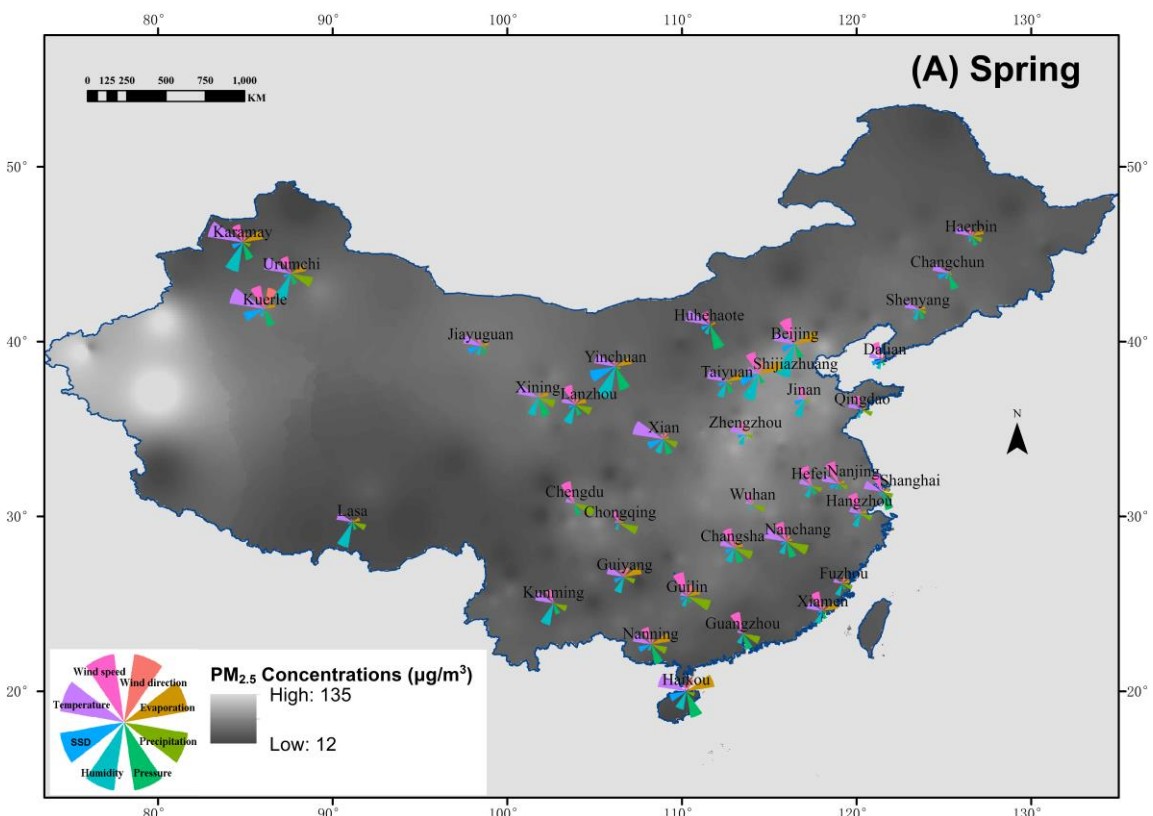


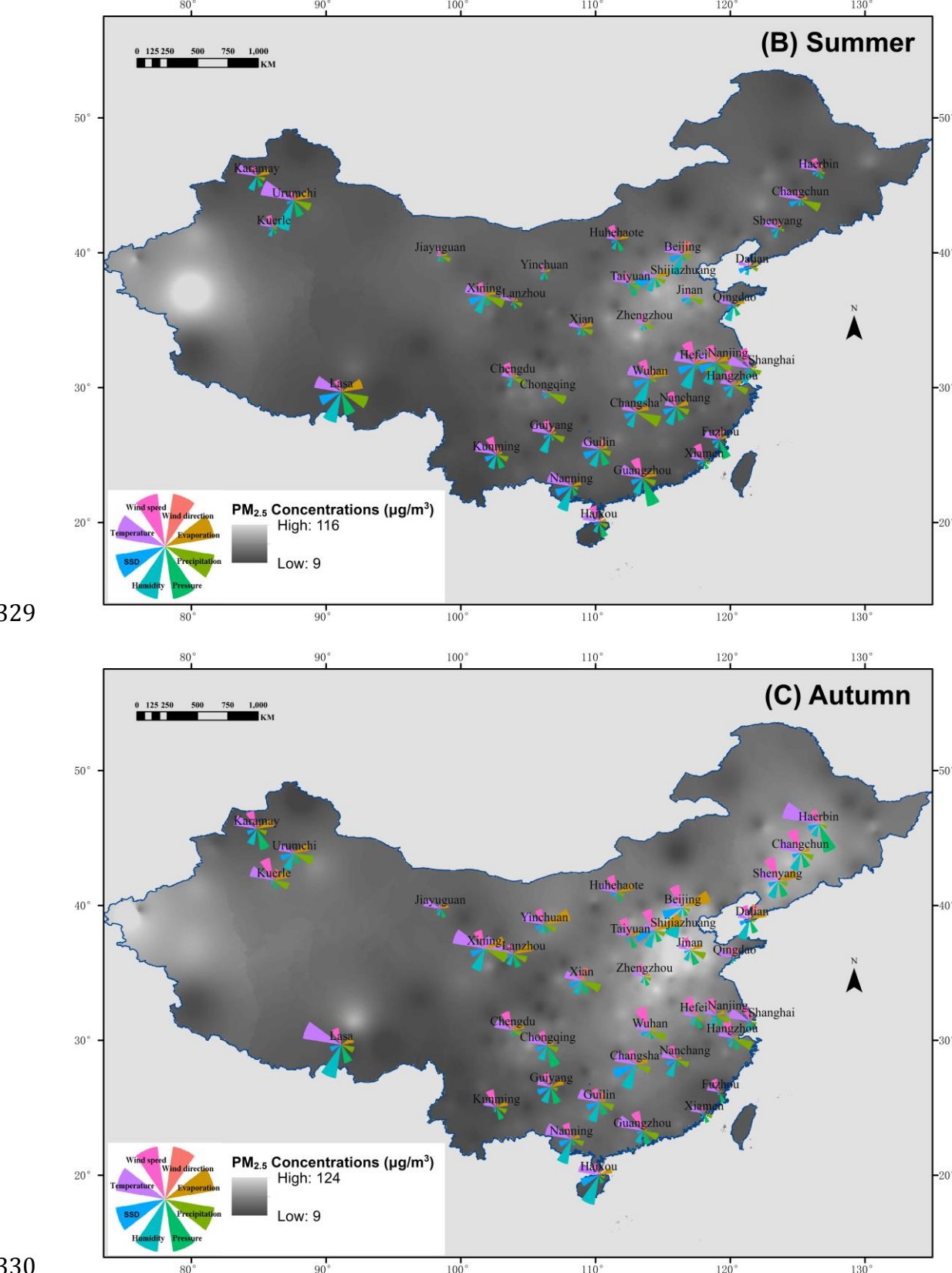




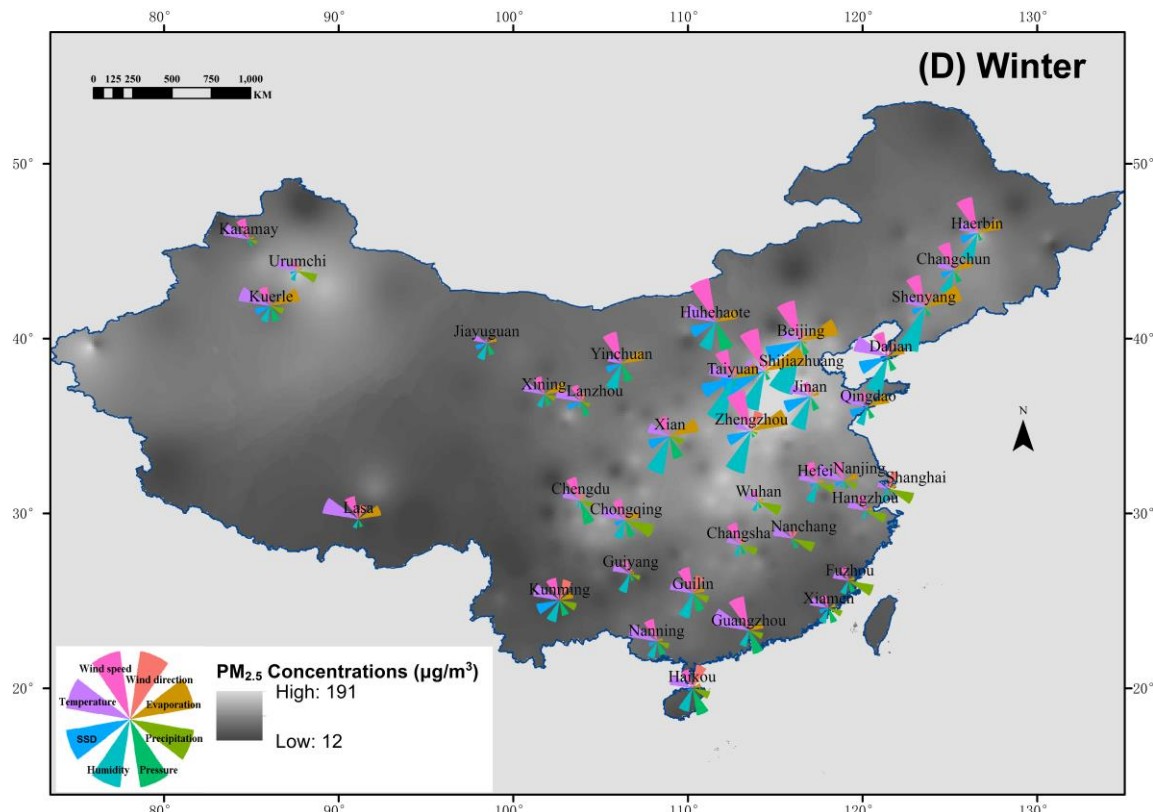

Fig 2. Wind rose map of influences of eight individual meteorological factors on PM$_{2.5}$
concentrations across China (37 representative cities) during 2014-2016


According to Fig 2, some spatial and temporal patterns of meteorological influences on
PM$_{2.5}$ concentrations at the national scale can be found as follows:

a. Like seasonal variations of PM$_{2.5}$ concentrations, the influences of individual
meteorological factors on local PM$_{2.5}$ concentrations vary significantly. For a specific city,
the dominant meteorological driver for PM$_{2.5}$ concentrations in one season may become
insignificant in another season. E.g. in winter, one major meteorological influencing
factor for Beijing is wind (The mean $\rho$ value during 2014-2016 was 0.57), which
exerts little influence on PM$_{2.5}$ concentrations in summer (The mean $\rho$ value during
2014-2016 was 0.10). Furthermore, it is noted that seasonal variations of meteorological
influences on PM$_{2.5}$ concentrations apply to all these representative cities, as the shape
and size of wind rose for each city change significantly across different seasons. Take
several mega cities in different regions for instance. During 2014-2016, the three major
meteorological influencing factors for PM$_{2.5}$ concentrations in Beijing, a mega city in the
North China plain, were as follows: humidity (0.48),wind (0.37) and evaporation (0.31)

for spring, humidity (0.39),temperature (0.34) and SSD (0.25) for summer, humidity (0.56),evaporation (0.51) and wind (0.41) for autumn, and humidity (0.76), wind (0.57) and evaporation (0.52) for winter. The three major meteorological influencing factors for PM$_{2.5}$ concentrations in Shanghai, a mega city in the Yangtze River Basin, were as follows: temperature (0.264), air pressure (0.260) and wind (0.25) for spring, temperature (0.40), wind (0.38) and humidity (0.27) for summer, temperature (0.39), wind (0.28) and humidity (0.17) for autumn, and precipitation (0.36), wind direction (0.25) and humidity (0.19) for winter. The three major meteorological influencing factors for PM$_{2.5}$ concentrations in Wuhan, a major city in Central China region, were as follows: precipitation ( 0.18), wind (0.16) and temperature (0.09) for spring, humidity (0.47), temperature (0.41) and wind (0.34) for summer, wind (0.44), precipitation (0.31) and temperature (0.26) for autumn, and precipitation (0.33), temperature (0.19) and wind (0.15) for winter. The three major meteorological influencing factors for PM$_{2.5}$ concentrations in Guangzhou, a major city in Southern China region, were as follows: wind (0.31), precipitation (0.24) and air pressure (0.23) for spring, air pressure (0.51), temperature (0.41) and wind (0.37) for summer, temperature (0.47), wind (0.36) and precipitation (0.29) for autumn, and temperature (0.52), wind (0.48) and air pressure (0.33). Notable seasonal variations of meteorological influences on PM$_{2.5}$ concentrations were found in these mega cities across China.b. In spite of notable differences in the shape and size of wind roses, meteorological influences on PM$_{2.5}$ concentrations cities are of some regional patterns. PM$_{2.5}$ concentrations in cities within the North China region are influenced by similar dominant meteorological factors, especially in winter, when PM$_{2.5}$ concentrations in these cities was high. Take four major cities, Beijing, Tianjin, Taiyuan and Shijiangzhuang, in the North China Plain for example. For winter, SSD, evaporation, humidity and wind were the major meteorological factors for PM$_{2.5}$ concentrations in the four cities and the $\rho$ value of these four factors was 0.50, 0.52, 0.76 and 0.57 for Beijing, 0.41, 0.44, 0.56 and 0.50 for Tianjin, 0.44, 0.36, 0.61 and 0.41 for Taiyuan, and 0.62, 0.58, 0.56 and 0.60 for Shijiazhuang respectively, presenting a similar regional pattern. Meanwhile, meteorological influences on PM$_{2.5}$ concentrations in cities within the Yangtze River Basin, especially the dominant factors, were also of some regional similarities. Take four major cities in the Yangtze River Basin, Shanghai, Nanjing, Hangzhou and Nanchang for example. For summer, precipitation, humidity, temperature and wind were the major meteorological factors for PM$_{2.5}$ concentrations in

these four cities and the $\rho$ value of these factors was 0.21, 0.27, 0.40 and 0.38 for
Shanghai, 0.29, 0.41, 0.34 and 0.33 for Nanjing, 0.28, 0.27, 0.23 and 0.27 for Hangzhou,
and 0.24, 0.33, 0.21 and 0.29 for Nanchang. Despite of some differences in the $\rho$
values, similar dominant meteorological factors and the similar magnitude of
meteorological influences demonstrated regional similarities of meteorological
influences on PM$_{2.5}$ concentrations in the Yangtze River Basin.
As we can see, meteorological influences on PM$_{2.5}$ concentrations in China are mainly
controlled by geographical conditions (e.g. terrain and landscape patterns).
c. For the heavily polluted North China region, the higher the local PM$_{2.5}$ concentrations,
the larger influence meteorological factors exerts on PM$_{2.5}$ concentrations. PM$_{2.5}$
concentrations are usually the highest in winter, causing serious air pollution episodes
across China, the North China region in particular. Meanwhile, PM$_{2.5}$ concentrations in
spring and summer are comparatively low. Accordingly, there are more influencing
meteorological factors on PM$_{2.5}$ concentrations for cities within this region and the $\rho$
value of these meteorological factors is notably larger in winter. Take four major cities in
the North China region for instance. For Beijing, the major influencing meteorological
factors in summer were temperature (0.34), humidity (0.39) and SSD (0.25) whilst the
major influencing meteorological factors in winter were humidity (0.76), wind (0.57),
evaporation (0.52) and SSD (0.5). For Tianjin, the major influencing meteorological
factors in summer were precipitation (0.34), temperature (0.22) and air pressure (0.25)
whilst the major influencing meteorological factors in winter were humidity (0.76),
wind (0.57), evaporation (0.52) and SSD (0.50). For Shijiazhuang, the major influencing
meteorological factors in summer were SSD (0.4), humidity (0.26) and evaporation (0.26)
whilst the major influencing meteorological factors in winter were SSD (0.62), wind
(0.60), evaporation (0.58) and humidity (0.56). For Taiyuan, the major influencing
meteorological factors in summer were temperature (0.32), air pressure (0.23) and
precipitation (0.20) whilst the major influencing meteorological factors in winter were
humidity (0.61), SSD (0.44) and wind (0.41). As explained, bidirectional interactions
between meteorological factors and PM$_{2.5}$ concentrations may lead to complicated
mechanisms that further enhance local PM$_{2.5}$ concentrations significantly. Therefore,
strong meteorological influences on PM$_{2.5}$ concentrations in winter are a major cause for
the form and persistence of high PM$_{2.5}$ concentrations within the North China region.

**4.2 Spatial and temporal variations of the dominant meteorological influence on**

**local PM₂.₅ concentrations across China**

Through statistical analysis, we selected the factor with the largest $\rho$ value as the dominant meteorological factor for local PM₂.₅ concentrations. The spatial and temporal variations of the dominant meteorological influence on local PM₂.₅ concentrations across China are demonstrated as Fig 3. According to Fig 3, some spatio-temporal characteristics of meteorological influences on PM₂.₅ concentrations can be further concluded:

a. The dominant meteorological factor for PM₂.₅ concentrations is closely related to geographical conditions. For instance, the factor of precipitation may exert a key influence on local PM₂.₅ concentrations in some coastal cities and cities within the Yangtze River Basin whilst this meteorological factor exerts limited influence on PM₂.₅ concentrations within some inland regions. Here we analyzed the $\rho$ value of precipitation in cities within the Yangtze River Basin and cities within the Beijing-Tianjin-Hebei region respectively. For winter, precipitation was the dominant factor for PM₂.₅ concentrations in Shanghai, Hangzhou and Nanchang within the Yangtze River Basin and the $\rho$ value of precipitation was 0.36, 0.29 and 0.31 respectively. Meanwhile, the $\rho$ value of precipitation in Beijing, Tianjin and Shijiazhuang within the Beijing-Tianjin-Hebei region was 0.08, 0.01 and 0.06 respectively.

b. Some meteorological factors can be the dominant factor for cities within different regions but some (e.g. evaporation and SSD) are mainly the dominant meteorological factor for PM₂.₅ concentrations in cities within some specific regions. In other words, some factors can be regarded as regional and national meteorological factors for PM₂.₅ concentrations, yet some meteorological factors are context-related influencing factors for local PM₂.₅ concentrations. Specifically, such factors as temperature, wind and humidity serve as the dominant meteorological factors in many regions, including Northeast, Northwest, coastal areas and inland areas; Meanwhile, such factors as SSD and wind direction serve as the dominant meteorological factors mainly in some inland regions. The prevalence of different meteorological factors across China can also be reflected according to the number of cities where this specific factor is the dominant factor for local PM₂.₅ concentrations. For winter, the number of cities with temperature, wind or humidity as the dominant factor was 56,48 and 44 respectively. Meanwhile, the

number of cities with SSD or wind direction as the dominant factor was 3 and
1respectively.
c. Similar to patterns revealed in Fig 2, the $\rho$ value for the dominant meteorological
factors is much larger in winter than that in summer. Furthermore, it is noted that the
dominant meteorological factors demonstrate more regional similarity in winter.
Specially, the dominant meteorological factors for $PM_{2.5}$ concentrations in the heavily
polluted North China region are more concentrated and homogeneously distributed in
winter (mainly the wind and humidity factor) whilst a diversity of dominant
meteorological factors (includes humidity, temperature, SSD and air pressure) for $PM_{2.5}$
concentrations is irregularly distributed within this region in summer. Take some major
cities in North China region for instance. For winter, the dominant meteorological factors
for   Beijing, Tianjin, Taiyuan, Zhangjiakou, Handan and Jining was humidity (0.76),
humidity (0.56), humidity (0.61), wind (0.62), humidity (0.43) and humidity (0.52)
respectively. Meanwhile, for summer, the dominant meteorological factors for Beijing,
Tianjin, Taiyuan, Zhangjiakou, Baoding, Handan and Jining was humidity (0.39),
precipitation (0.28), temperature (0.23), temperature (0.47), air pressure (0.21) and SSD
(0.18). According to this pattern, when a regional $PM_{2.5}$-induced air pollution episode
occurs in winter, the regional air quality is more likely to be simultaneously improved by
the same meteorological factor. This is consistent with the common scene in winter that
regional air pollution episodes in the Beijing-Tianjin-Hebei region can be considerably
mitigated by strong northwesterly synoptic winds,  which are produced by presence of
high air pressure in northwest Beijing (NW-High) (Tie et al., 2015; Miao et al., 2015).
On the other hand, regional air pollution in summer can hardly be solved simultaneously
through one specific meteorological factor.

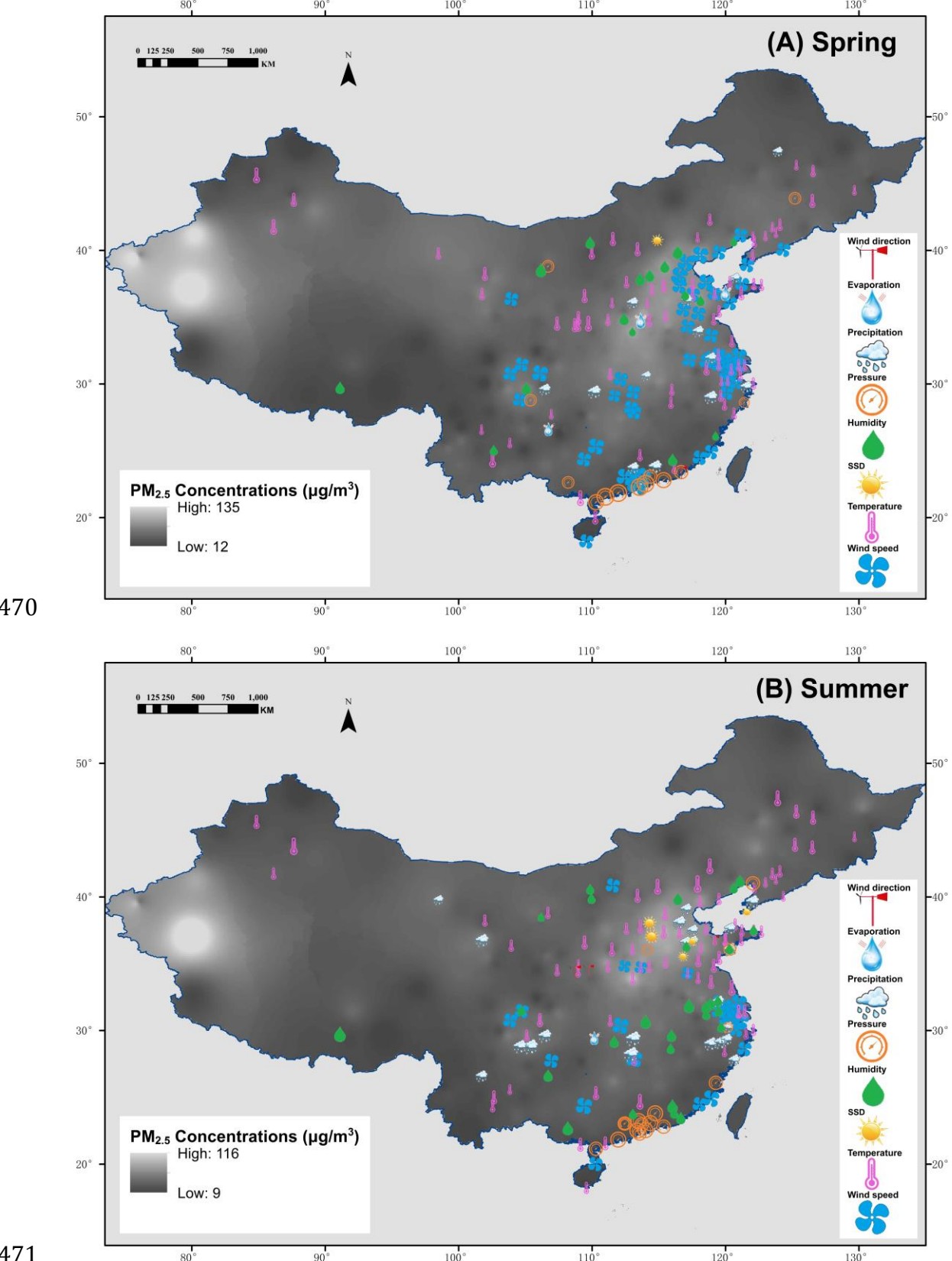



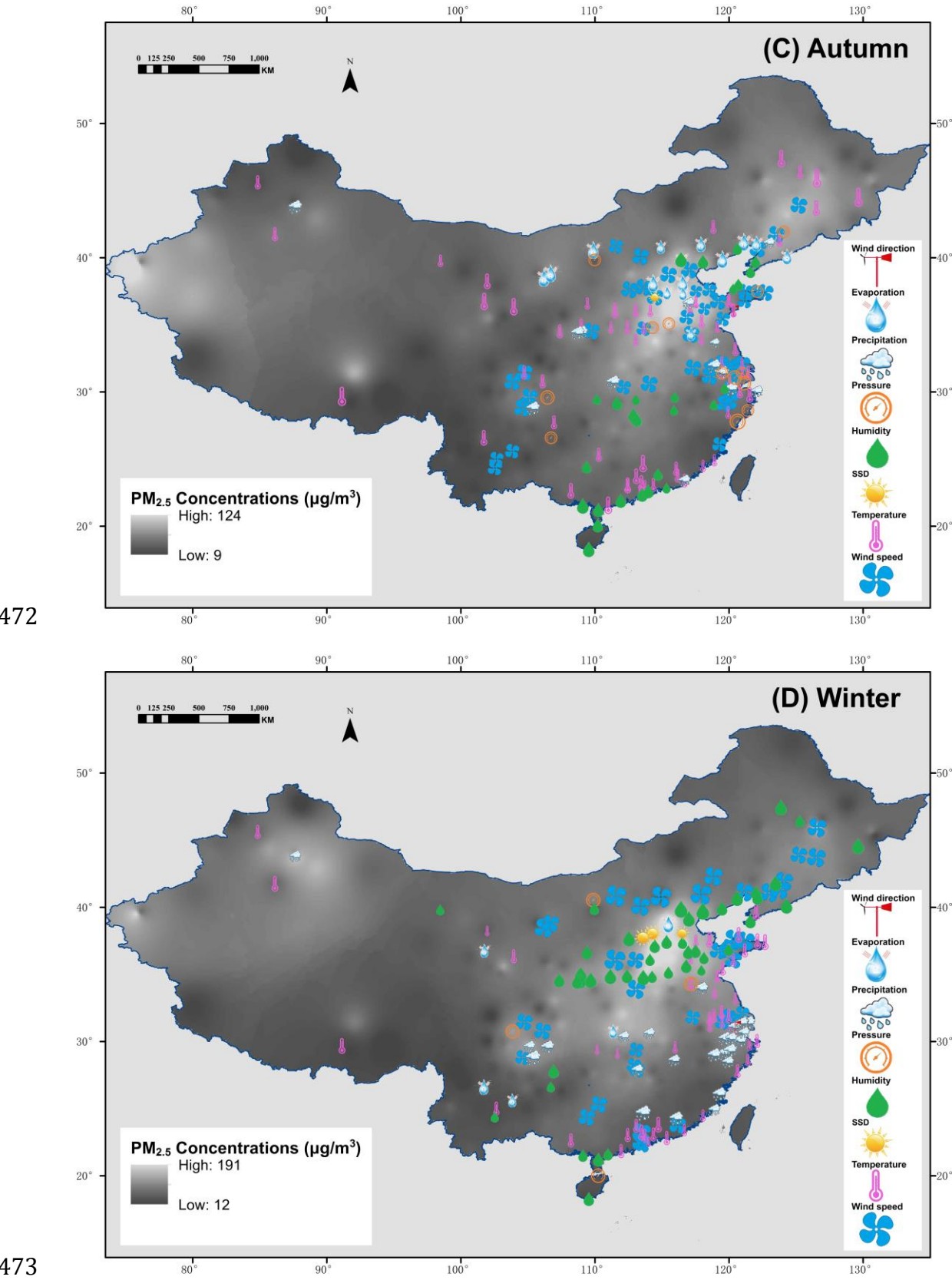

**Fig 3. The dominant meteorological factor for local PM$_{2.5}$ concentrations in 188 monitoring cities across China**

The size of symbols indicates the $\rho$ value of the meteorological factor on local PM$_{2.5}$ concentrations.

**4.3 Comparative statistics of the influence of individual meteorological factors on**
**local PM$_{2.5}$ concentrations across China**
In addition to meteorological influences on PM$_{2.5}$ concentrations for individual cities,
we examined and compared the comprehensive influence of individual meteorological
factors on PM$_{2.5}$ concentrations at a national scale. The results are presented as Table
1 and Fig 4.
**Table 1. The comparison of the influence of individual meteorological factors on**
**PM$_{2.5}$ concentrations in 188 cities across China (2014-2016)**

| Season | Factor | TEM | SSD | PRE | EVP | PRS | RHU | WIN | Dir_WIN |
|--------|--------|-----|-----|-----|-----|-----|-----|-----|---------|
| **Spring** | **No. of cities[1]** | 76 | 1 | 13 | 3 | 13 | 17 | 64 | 1 |
| | Mean $\rho$ value | 0.254 | 0.102 | 0.143 | 0.108 | 0.177 | 0.161 | 0.222 | 0.094 |
| | SD of $\rho$ value | 0.106 | 0.071 | 0.088 | 0.081 | 0.123 | 0.105 | 0.102 | 0.077 |
| | Max $\rho$ value | 0.572 | 0.366 | 0.385 | 0.397 | 0.653 | 0.475 | 0.595 | 0.429 |
| **Summer** | No. of cities | 78 | 5 | 22 | 1 | 20 | 32 | 27 | 3 |
| | Mean $\rho$ value | 0.272 | 0.136 | 0.183 | 0.137 | 0.163 | 0.219 | 0.191 | 0.087 |
| | SD of $\rho$ value | 0.098 | 0.086 | 0.099 | 0.088 | 0.109 | 0.118 | 0.095 | 0.062 |
| | Max $\rho$ value | 0.604 | 0.433 | 0.536 | 0.399 | 0.518 | 0.562 | 0.453 | 0.311 |
| **Autumn** | No. of cities | 70 | 1 | 13 | 15 | 13 | 27 | 48 | 1 |
| | Mean $\rho$ value | 0.316 | 0.164 | 0.191 | 0.181 | 0.199 | 0.247 | 0.265 | 0.104 |
| | SD of $\rho$ value | 0.109 | 0.098 | 0.093 | 0.117 | 0.091 | 0.125 | 0.089 | 0.074 |
| | Max $\rho$ value | 0.702 | 0.479 | 0.430 | 0.514 | 0.524 | 0.662 | 0.488 | 0.331 |
| **Winter** | No. of cities | 56 | 3 | 27 | 5 | 4 | 48 | 44 | 1 |
| | Mean $\rho$ value | 0.306 | 0.183 | 0.166 | 0.190 | 0.180 | 0.304 | 0.299 | 0.119 |
| | SD of $\rho$ value | 0.094 | 0.129 | 0.115 | 0.130 | 0.086 | 0.161 | 0.136 | 0.092 |
| | Max $\rho$ value | 0.527 | 0.615 | 0.473 | 0.595 | 0.427 | 0.755 | 0.623 | 0.560 |

[1]No. of cities: the number of cities with this factor as the dominant meteorological factor (its $\rho$ value
is the largest amongst eight factors) on local PM$_{2.5}$ concentrations.

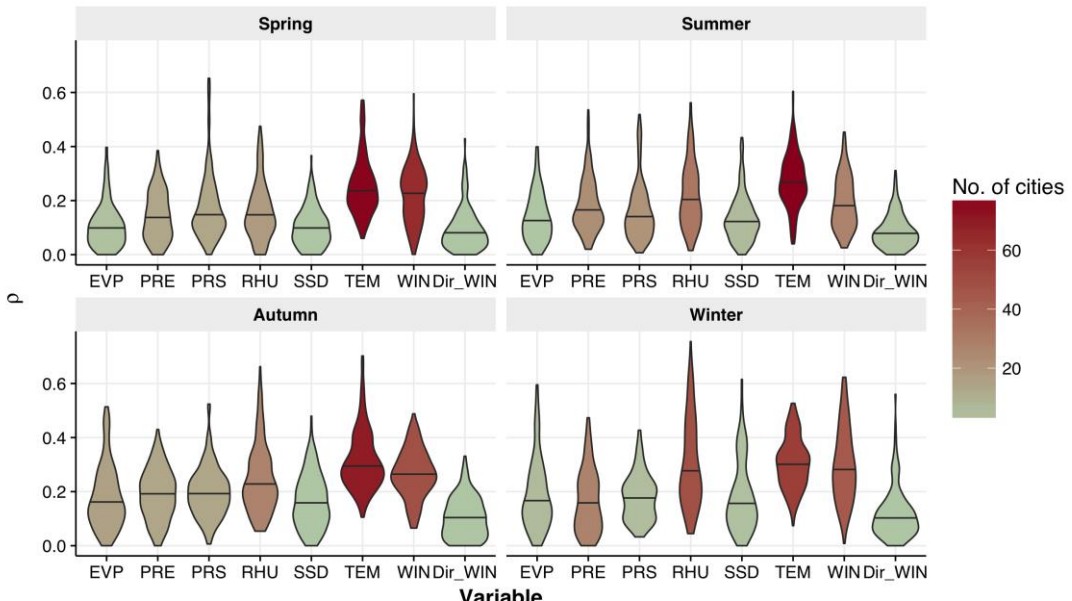


**Fig 4. Violin plots of the influence of eight different meteorological factors on local PM$_{2.5}$ concentrations in 188 cities across China**

No. of cities: the number of cities with this factor as the dominant meteorological factor (its $\rho$ value is the largest amongst eight factors) on local PM$_{2.5}$ concentrations. The shape of the violin bars indicated the frequency distribution of $\rho$ value for 188 cities.

We compared the influence of individual meteorological factors on PM$_{2.5}$ concentrations from different perspectives.

a. From a national perspective, temperature, humidity, and wind exert stronger influences on local PM$_{2.5}$ concentrations than other factors. The annual mean $\rho$ value for temperature, wind and humidity was 0.287, 0.244 and 0.233, compared with wind direction (0.101), SSD (0.146), evaporation (0.155), precipitation (0.171) and air pressure (0.180). Amongst the eight factors, temperature was found to be the most influential meteorological factor for general PM$_{2.5}$ concentrations in China. In addition to the largest mean $\rho$ value, temperature was the dominant meteorological factors for most cities in all seasons. Furthermore, the Coefficient of Variation (SD /mean×100%) for temperature was much smaller than other factors, indicating the consistent influence of temperature on local PM$_{2.5}$ concentrations across China.

b. Although some meteorological factors exert a limited influence on PM$_{2.5}$ concentrations at a national scale, these factors may be a key meteorological factor for local PM$_{2.5}$ concentrations. As shown in Table 1, the max $\rho$ value for each

meteorological factor was large than 0.35 for all seasons (except for the wind
direction factor in summer and autumn), indicating a very strong influence on local
$PM_{2.5}$ concentrations in some specific regions. As a result, when analyzing
meteorological influences on local $PM_{2.5}$ concentrations for a specific city,
meteorological factors that have little influence on $PM_{2.5}$ concentrations at a large
scale should also be comprehensively considered.
c. Some factors (e.g. precipitation in summer and winter) may be the dominant
meteorological factors for a large number of cities, though the mean $\rho$ value
remained small. This may be attributed to the fact that these meteorological factors
mainly exert influence on local $PM_{2.5}$ concentrations in those cities (seasons) where
(when) the general $PM_{2.5}$ concentrations is not high. Taking the precipitation as an
example. Luo et al. (2017). pointed out that there may be thresholds for the negative
influences of precipitations on PM2.5 concentrations and Guo et al. (2016) found that
the same amount of precipitation led to a weaker washing-off effect in areas with
higher $PM_{2.5}$ concentrations. Hence, precipitation mainly exerts a dominant influence
on local $PM_{2.5}$ concentrations in winter for Yangtze River Basin or coastal cities,
where the amount of precipitation is large and the $PM_{2.5}$ concentration is low, whilst
precipitation exerts a limited role in northern China, where the amount of
precipitation is small and the $PM_{2.5}$ concentration is high. Therefore, as explained
above, comprehensive meteorological influences on $PM_{2.5}$ concentrations are limited
considerably.
**5 Discussion**
Despite the lack of a comprehensive comparison of meteorological influences on
$PM_{2.5}$ concentrations across different regions, correlations between individual
meteorological factors and $PM_{2.5}$ concentrations have been analyzed in such mega
cities as Nanjing ( Chen, T. et al, 2016; Shen and Li., 2016;), Beijing (Huang et al,
2015; Yin et al., 2016), Wuhan ( Zhang et al, 2017), Hangzhou (Jian et al., 2012),
Chengdu ( Zeng and Zhang, et al. 2017) and Hong Kong (Fung et al, 2014). These
studies mainly employed correlation analysis to quantify the influence of several
meteorological factors on $PM_{2.5}$ concentrations and suggested that meteorological
influences on $PM_{2.5}$ concentrations varied significantly across regions. The
dominant meteorological factors for $P_{2.5}$ concentrations (presented as the largest
correlation coefficients in previous studies and the largest $\rho$ value in this research)
demonstrated notable regional differences. For Nanjing (Chen, T. et al., 2016), a
mega city in the Yangtze River, and Hong Kong (Fung et al., 2014), a mega coastal
city, precipitation exerted the strongest influence whilst wind speed exerted a
weak influence on $PM_{2.5}$ concentrations in winter. On the other hand, for winter,
wind speed was the dominant meteorological factor for $PM_{2.5}$ concentrations in
Beijing (Huang et al., 2015.) , a mega city in North China, and precipitation played
a weak role in affecting local $PM_{2.5}$ concentrations . These studies generally
analyzed and compared the influences of different meteorological factors on $PM_{2.5}$
concentrations and extracted the dominant meteorological influencing factors for
specific areas. Compared with studies at a local or regional scale, this research
conducted at the national scale provided a better understanding of spatial and
temporal patterns of meteorological influences on $PM_{2.,5}$ concentrations across China,
for the following reasons. a. A national perspective. Previous studies conducted at a
local scale mainly focused on a specific city (e.g. Beijing), and can hardly reveal
spatio-temporal patterns of meteorological influences on $PM_{2.5}$ concentrations at a
large scale (e.g. the North China plain). This research, on the other hand, quantified
the influence of meteorological factors on $PM_{2.5}$ concentrations for 188 cities across
China, and thus revealed some regional patterns of meteorological influences on
$PM_{2.5}$ concentrations in some typical regions (e.g. North China region or Yangtze
River Basin). b. A unified research period and set of meteorological factors. Previous
studies employed short-term observation data (e.g. one season or one year) to
examine the meteorological influences on local $PM_{2.5}$ concentrations in specific cities.
Due to the discrepancy in research periods and sets of meteorological factors, the
findings from different local-scale studies cannot be compared and comprehensively
understood. This research employed daily $PM_{2.5}$ and meteorological data of three
consecutive years and a unified set of eight meteorological factors for all 188
monitoring cities and thus meteorological influences on $PM_{2.5}$ concentrations across
China can be effectively compared without significant influences from inter-annual
variations. c. A robust causality analysis method. Due to complicated interactions
between different meteorological factors, correlations analysis, as introduced above,
may lead to large bias in quantifying the meteorological influences on $PM_{2.5}$
concentrations. Similarly, the correlation coefficient between individual
meteorological factors and $PM_{2.5}$ concentrations cannot be used as a reliable indicator
to compare quantitative influences of individual meteorological factors on $PM_{2.5}$
across different cities. This research employed a robust CCM method, which removes
the influence of other factors, and effectively quantified the coupling between $PM_{2.5}$
concentrations and a set of meteorological factors. The $\rho$ value of each
meteorological factor on $PM_{2.5}$ concentration can be compared between different
cities. Based on national statistics across China, this research concluded that the
influence of temperature, humidity and wind, especially temperature, on $PM_{2.5}$
concentrations was much larger than that of other meteorological factors, which could
not be revealed by previous local and regional scale studies.

The findings from this research were consistent with and a major extension of those
from previous studies by quantifying the influence of individual meteorological
factors in a large number of cities across China using a more robust causality analysis
method. Similar to previous studies, this study also revealed notable differences in
meteorological influences on $PM_{2.5}$ concentrations at the national scale, the major
reason for which was different meteorological conditions and complicated
mechanisms of $PM_{2.5}$-meteorology interactions. Firstly, notable differences existed in
meteorological conditions across China. For instance, in winter, the frequency and
intensity of precipitation are much higher and stronger in coastal areas than those in
the North China region, where the frequency of strong winds is high in winter.
Therefore, precipitation exerts a large influence on $PM_{2.5}$ concentrations in coastal
regions whilst wind is the key influencing factor for $PM_{2.5}$ concentrations in the North
China region in winter. Secondly, in addition to the large variations in the values of
correlation coefficients, the interaction mechanisms between individual
meteorological factors and $PM_{2.5}$ concentrations may also vary significantly across
regions. For such meteorological influences as wind speed, its negative effect on
$PM_{2.5}$ concentrations was consistent in China (He e al., 2017). On the other hand, He
et al. (2017) suggested that temperature and humidity were either positively or
negatively correlated with $PM_{2.5}$ concentrations in different regions of China. In terms
of humidity, when the humidity is low, $PM_{2.5}$ concentration increases with the increase
of humidity due to hygroscopic increase and accumulation of $PM_{2.5}$ (Fu et al., 2016).
When the humidity continues to grow, the particles grow too heavy to stay in the air,
leading to dry (particles drop to the ground) (Wang, J., & Ogawa, S. (2015)) and wet
deposition (precipitation) (Li et al., 2015b), and the reduction of $PM_{2.5}$ concentrations.
Similarly, there may be thresholds for the negative influences of precipitations on
$PM_{2.5}$ concentrations (Luo et al., 2017). Heavy precipitation can have a strong
washing-off effect on $PM_{2.5}$ concentrations and notably reduce PM2.5 concentrations.
Meanwhile, slight precipitation may not effectively remove the high-concentration
$PM_{2.5}$. Instead, the slight precipitation may induce enhanced relative humidity and
thus lead to the increase of $PM_{2.5}$ concentrations. Meanwhile, the washing-off effect
from the same amount of precipitation on $PM_{2.5}$ concentrations in Xi'an, a city with
higher $PM_{2.5}$ concentrations, was lower than that in Guangzhou (Guo et al., 2016),
indicating local $PM_{2.5}$ concentrations also exerted a key role in the negative effects of
precipitation. Meanwhile,temperature can either be negatively correlated with $PM_{2.5}$
concentrations by accelerating the flow circulation and promoting the dispersion of
$PM_{2.5}$ (Li et al., 2015b), or positively correlated with $PM_{2.5}$ concentrations through
inversion events (Jian et al., 2012). Given the complexity of interactions between
meteorological factors and $PM_{2.5}$, characteristics and variations of influences of
individual meteorological factors on $PM_{2.5}$ concentrations should be further
investigated for specific regions across China respectively based on long-term
observation data.
Due to highly complicated atmospheric environment and the difficulty in acquiring
true data of exhaust emission, commonly used models for air quality prediction(e.g.
CAMx, CMAQ and WRFCHEM) may lead to large biases and uncertainty when
applied to China. On the other hand, without prior knowledge of mechanisms of high
$PM_{2.5}$ concentrations and information of exhaust emission, statistical models can
achieve satisfactory forecasting results based on massive historical data (Cheng et al.,
2015). Compared with the static models, dynamic statistical models additionally
consider the meteorological influences on $PM_{2.5}$ concentrations and some
meteorological factors that are of stable, representative and strong correlations with
$PM_{2.5}$ are selected for forecasting $PM_{2.5}$ concentrations. Meanwhile, many recent
studies (Cheng et al., 2017; Guo et al., 2017; Lu et al., 2017; Ni et al. 2017; etc) have
recognized the meteorological influences on the evolution of $PM_{2.5}$ concentrations and
included some key meteorological factors in their models for $PM_{2.5}$ estimation.
However, most $PM_{2.5}$ estimation and forecasting models mainly employed correlation
analysis to reveal the influence of individual meteorological factors on PM$_{2.5}$
concentrations. Due to complicated interactions in atmospheric environment, the
correlation coefficient between meteorological factors and PM$_{2.5}$ concentrations is
usually much larger than the $\rho$ value and overestimates the influence of individual
meteorological factors on PM$_{2.5}$ concentrations. In this case, this research provides
useful reference for improving existing statistical models. By incorporating the
$\rho$ value, instead of the correlation coefficient, of different factors into corresponding
GAM (Generalized Additive Models) and adjusting parameters accordingly, we may
significantly improve the reliability of future estimation and forecasting of PM$_{2.5}$
concentrations.
Quantified causality of individual meteorological factors on PM$_{2.5}$ concentrations
provides useful decision support for evaluating relevant environmental projects,
which aim to improve local and regional air quality through meteorological means
Specifically, a forthcoming Beijing wind-corridor project
(http://www.bj.xinhuanet.com/bjyw/yqphb/2016-05/16/c_1118870801.htm) has
become a hot social and scientific issue. Herein, our research suggests that wind is a
dominant meteorological factor for winter PM$_{2.5}$ concentrations in Beijing and can
significantly influence PM$_{2.5}$ concentrations through direct and indirect
mechanisms( Chen,Z. et al., 2017). In consequence, the wind-corridor project may
directly allow in more strong wind, which thus leads to a larger value of SSD and
EVP and a smaller value of RHU. The change of SSD, RHU and EVP values can
further induce the reduction of PM$_{2.5}$ concentrations. From this perspective, the
Beijing wind-corridor project has good potential to improve local and regional air
quality. In addition to the wind-corridor project, some scholars and decision makers
have proposed other meteorological means for reducing PM$_{2.5}$ concentrations. For
instance, Yu (2014) suggested that water spraying from high buildings and water
towers in urban areas was an efficient way to reduce PM$_{2.5}$ concentrations rapidly by
simulating the process of precipitation. However, some limitations, such as the
humidity control and potential icing risk, remained. In the near future, with growing
attention on the improvement of air quality, more environmental projects should be
properly designed and implemented. According to this research, meteorological
influences on PM$_{2.5}$ concentrations vary notably across China. Given the diversity of
dominant meteorological factors on local $PM_{2.5}$ concentrations in different regions
and seasons, it is more efficient to design meteorological means accordingly. For the
heavily polluted North China region, especially the Beijing-Tianjin-Hebei region, the
northwesterly synoptic wind(Tie et al., 2015; Miao et al., 2015)is much stronger in
winter than winds in summer and exerts a dominant influence on $PM_{2.5}$ concentrations
(Chen et al., 2017). Furthermore, in North China, the $PM_{2.5}$ concentration is much
more sensitive to the change of wind speed than that of other meteorological factors
(Gao et al., 2016). Meanwhile, wind-speed induced climate change led to the change
of $PM_{2.5}$ concentrations by as much as 12.0 $\mu gm^{-3}$, compared with the change of
$PM_{2.5}$ concentrations by up to 4.0 $\mu gm^{-3}$ in south-eastern, northwestern and
south-western China (Tai et al., 2010). Considering the strong winds in winter, the
dominant influence of wind speed on $PM_{2.5}$ concentrations and the sensitivity of
$PM_{2.5}$ feedbacks to the change of wind speed, meteorological means for encouraging
strong winds are more likely to reduce $PM_{2.5}$ concentrations considerably in North
China. Similarly, Luo et al. (2017) suggested that only precipitation with a certain
magnitude can lead to the washing-off effect of $PM_{2.5}$ concentrations whilst Guo et al.
(2016) revealed that the variation of $PM_{2.5}$ concentrations was more sensitive to the
same amount of precipitation in areas with lower $PM_{2.5}$ concentrations. Therefore,
meteorological means for inducing precipitation are more likely to improve air quality
in coastal cities and cities within the Yangtze River basin, where there is a large
amount of precipitation and relatively low $PM_{2.5}$ concentrations.
**6 Conclusions**
Previous studies examined the correlation between individual meteorological
influences and $PM_{2.5}$ concentrations in some specific cities and the comparison
between these studies indicated that meteorological influences on $PM_{2.5}$
concentrations varied significantly across cities and seasons. However, these scattered
studies conducted at the local scale cannot reveal regional patterns of meteorological
influences on $PM_{2.5}$ concentrations. Furthermore, previous studies generally selected
different research periods and meteorological factors, making the comparison of
findings from different studies less robust. Thirdly, these studies employed the
correlation analysis, which may be biased significantly due to the complicated
interactions between individual meteorological factors. This research is a major
extension of previous studies.   Based on a robust causality analysis method CCM,
we quantified and compared the influence of eight meteorological factors on local
$PM_{2.5}$ concentrations for 188 monitoring cities across China using $PM_{2.5}$ and
meteorological observation data from 2014.3 to 2017.2. Similar to previous studies
conducted at the local scale, this research further indicated that meteorological
influences on $PM_{2.5}$ concentrations were of notable seasonal and spatial variations at
the national scale. Furthermore, this research revealed some regional patterns and
comprehensive statistics of the influence of individual meteorological factors on
$PM_{2.5}$ concentrations, which cannot be understood through small-scale case studies.
For the heavily polluted North China region, the higher $PM_{2.5}$ concentrations, the
stronger influence meteorological factors exert on local $PM_{2.5}$ concentrations. The
dominant meteorological factor for $PM_{2.5}$ concentrations is closely related to
geographical conditions. For heavily polluted winter, precipitation exerts a key
influence on local $PM_{2.5}$ concentrations in most coastal areas and the Yangtze River
basin, whilst the dominant meteorological driver for $PM_{2.5}$ concentrations is wind in
the North China regions. At the national scale, the influence of temperature, humidity
and wind on local $PM_{2.5}$ concentrations is much larger than that of other factors, and
*temperature* exerts the strongest and most stable influences on national $PM_{2.5}$
concentrations in all seasons. The influence of individual meteorological factors on
$PM_{2.5}$ concentrations extracted in this research provides more reliable reference for
better modelling and forecasting local and regional $PM_{2.5}$ concentrations. Given the
significant variations of meteorological influences on $PM_{2.5}$ concentrations across
China, environmental projects aiming for improving local air quality should be
designed and implemented accordingly.
**Acknowledgement**
This research is supported by National Natural Science Foundation of China (Grant
Nos. 210100066), the National Key Research and Development Program of China
(NO.2016YFA0600104), the Fundamental Research Funds for the Central
Universities, Ministry of Environmental Protection (201409005) and Beijing Training
Support Project for excellent scholars (2015000020124G059).

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
