# Peer review of "Understanding meteorological influences on PM$_{2.5}$ concentrations across China: a temporal and spatial perspective"

_Atmospheric Chemistry and Physics, 2017_

## Referee Comment (RC1) · Anonymous Referee #3 · 21 Aug 2017

General Comments

This is an interesting paper that applies an exciting and fairly new statistical method (convergent cross mapping) to quantify the relationship between local air quality and local meteorology (wind, temperature, precipitation, etc.). The authors argue that unlike a simple correlation analysis, this method is able to demonstrate causal relationships between variables. My understanding is that this method is quite new, and it is central to this study, so I think this paper would benefit from a clearer discussion of why it is better than correlation at determining causal relationships. Would the authors' findings have differed significantly if they used correlation instead of CCM?

[Figure]

There are some aspects of CCM that were not clear to me. Does CCM account for relationships between meteorological factors? E.g. if wind is affecting both precipitation and PM, how is the affect of wind on precip on PM counted? Another question I have is what produces a large value. For example, are the values higher in winter simply because there is more PM available to be effected? Or as another example, is precipitation more effective at removing PM along the coasts because it rains more? If there were a way to normalize by the amount of total rainfall, would precip still be more important along the coasts than in the drier interior?

My second major concern is that this study uses a single year of data to make general comments about PM-meteorology relationships. This gives us little sense of how much year-to-year variability may exist in these relationships and generally weakens the conclusions. If CCM is too computationally expensive to use on multiple years, perhaps a different method could be used to supplement it.

Finally, I would like to see a deeper discussion of the scientific significance of this work. As it is written currently, this paper is almost purely descriptive. The new method is interesting, but the paper could do a better job of articulating what we are learning from it. Perhaps some discussion about why different meteorological factors are more/less important in different regions/seasons, for example, would help give the paper more depth. I would also suggest spending more time discussing the implications for modeling, especially in the introduction and conclusions, as that was what I took away as the most important implication in this paper.

Specific Comments

pp 3, ln 93-95 - Can you elaborate further on how your previous study showed that CCM is better than correlation for the benefit of readers who have not read that paper. It is important for your results here to make as clear as possible why CCM is a better method/provides new insights.

pp 8 ln 221 – Are the results sensitive to the choice of parameters?

pp 9 ln 237 – how is the value of determined? Are you calculating the limit or taking the value at a specific time series length? What about cases such as PM2.5 xmap minTEM, which (at least by eye) does not appear to be converging? Wouldn't that suggest that minTEM was not influencing PM2.5 at all?

pp 9 ln 244 – It's not clear here how PM is changing wind speed

pp 10 ln 270 - I would guess that (out of the 189 cities) those clustered regionally would show similar maps. Is that true? I.e. are the 37 cities shown representative of the cities not shown?

pp 13 ln 292-296 – This only seems to be true in some seasons.

pp 13 ln 299-300 - Can you quantify this more rigorously? By eye, it seems like there are enough outliers to call this in to question

pp 14 ln 337-339 – This seems true for winter vs. summer, but what about spring vs autumn?

pp 18 Fig 4 caption – Is there a particular argument for only including the dominant factor in each city?

Section 5.1 of the discussion feels out of place. This is more of a discussion of what we already know about aerosol-meteorology interactions than a discussion of the implications of the work done in this study. I would recommend either rewriting it so that it builds more on the results from this paper, or cut it and integrate the important information into earlier sections.

pp 24 ln 562-566 – as per my earlier comment, I am not sure that you have shown this. If Beijing were to receive the same amount of precipitation that a coastal city does, is it possible that precipitation would become more important in Beijing? Does looking at how a specific factor changed PM in a year tell us about how effective changes in that factor would be?

Technical Corrections

pp 1, ln 17 (and later occurrences) - "causality influence" is redundant. "Influence" already implies causality.

pp 2, ln 35-36 - "Amongst these environmental elements, . . . concerned social and ecological issues." The wording of this sentence is unclear.

pp 2, ln 42 - "Serious haze not only influences peoples daily life," this wording is vague. How does haze influence peoples daily lives?

pp 2, ln 57 - "controversy" should be changed to "controversial"

pp 3, ln 68 - ". . . fractions of three different sizes. . ." this is unclear. Authors should indicate that they are talking about aerosols, and specify the sizes.

pp 3, ln 7 - what region is this study referring to?

pp 3, ln 86 - word choice: I would suggest "well studied" instead of "massively studied"

pp 4, ln 119 – what does API stand for?

pp 5, ln 136-137 – I'm not clear on what small and large evaporation are.

pp 5, ln 144 – sunshine duration for the day is a less widely used term and should be defined here

pp 5 ln 146 – what qualifies as extreme wind speed?

pp 5 ln 146 – how is max wind direction defined?

pp 6 ln 174 – "Two time series" is unnecessary. Suggest changing the sentence to "{X}=[X(1),. . .X(L)] and {Y}=[Y(1),. . .Y(L)] are defined as the temporal variations of variables X and Y."

pp 6 ln 175 – It's unclear what r and S are.

pp 8 ln 217 – why can E be 2 or 3?

pp 14 ln 331-332 – the phrasing here is unclear.

pp 18 ln 374-376 – is there a way to test if these values are significant?

pp 20 ln 419 – this wording is unclear

pp 20 ln 426 – Wikipedia is not an appropriate source. Better to cite a scientific paper that defines SSD.

pp 20 ln 440 – Temperature inversion is certainly important, but none of the metrics in this study measure it directly.

pp 21 ln 447 – what about horizontal transport (advection)?

pp 21 ln 446 – change "social economic" to "socio-economic"

pp 22 ln 492 – change "negative causality on" to "decreases" and "positive causality on" to "increases"

pp 23 ln 527-528 – do you have citations for this?

pp 24 ln 544 – can you give more details/citations about the controversy?

---

## Referee Comment (RC2) · Anonymous Referee #1 · 21 Aug 2017

Chen et al. present an interesting analysis of the spatial and temporal variation of the relationships of meteorology and PM2.5 over most of China. The cross convergent mapping analysis provides a unique method of understanding the causality of the relationships, which might otherwise be missed with typical correlation analysis. They highlight that the meteorological influence on PM2.5 varies widely by location and season, and that attempts to engineer favorable air quality meteorology should take these differences into account. The paper is well-written and relatively thorough, however it requires some additional explanations and detail. Thus I recommend publication following minor revision.

Page 2, lines 56-57: "Although quantitative contributions of different sources (e.g. coal burning and automobile exhaust) to airborne pollutants remain controversy" – It's not clear what you mean here with the "controversial" – politically or scientifically? If scientifically – the direct emissions and/or subsequent chemistry?

Page 4, section 2.1.1: How do you quality control the data and/or deal with missing data?

CCM method: How does the time lag parameter affect the results? The resolution of the map is mentioned but how does it affect the physical interpretation of the results? – Especially for those variables that may act on a shorter time scale.

Page 13, lines 295-296: This causality seems to be backwards: i.e., why would differences in PM levels cause differences in meteorological influences? What mechanism would cause this?

Page 14, lines 330-333: This sentence is very vague – can you be more specific?

Figure 2 and 3: I would suggest the background of concentrations be in a gray scale so the colored icons/wind roses stand out more. Also, how different would the maps be if the correlation coefficient were used instead? A statement or two would reinforce the argument for the use of CCM rather than correlation coefficient.

Page 20, lines 414-420: While higher relative humidity does lead to hygroscopic growth of aerosols, this is probably not evident in the observed concentrations since most measurements are taken at a constant relative humidity (e.g, 35% in US and Europe). Measurements in China may not do this, and if so, should be explicitly stated since this can have a major effect to aerosol mass depending on the composition of the aerosol.

Page 23, lines 515-535: This paragraph seems out of place with the rest of the section.

Page 23, line 525: I am not able to read Cheng et al. (2015), but I'm wondering what the model is using for predictors? If they are "static" models, isn't that just the mean state? I'm having a hard time understanding. If the argument is to use CCM instead of

correlations, an example (see above) would help to reinforce this.

Page 24, lines 562-566: How does the frequency of precipitation affect this statement? For example, if precipitation is rare in Beijing during winter, especially compared to the Yangtze River Basin.

Results/Discussion: Much of this review of meteorology-PM2.5 relationships in the discussion would probably be better suited in the introduction and in the results as it pertains to different locations within China. Many of the statements in the results are rather vague (e.g., page 14, line 330-333) and could be elaborated to include specific meteorological factors and specific locations.

Minor comments

Page 2, line 61: were correlated

Page 3, line 68: "fractions of three different sizes" of particulate matter

Page 4, lines 119-120: What does this sentence mean?

Page 12, line 288: Awkward wording

Page 20, line 426: Wikipedia is not an appropriate citation.

Page 20, line 427 and elsewhere: Check your usage of "by analogy" – you may be looking for a different phrase.

Page 20, line 433 and elsewhere: Check subject verb agreement, specifically for "PM2.5" and "concentration(s)"

---

## Referee Comment (RC3) · Anonymous Referee #2 · 22 Aug 2017

This paper attempts to investigate the meteorological influence on PM2.5 concentrations in China at the national scale using a convergent cross-mapping (CCM) method. This method is somewhat new to the atmospheric chemistry community, but the physical mechanism as discussed in this paper is very descriptive and already well-known. Overall I don't feel these results are significant enough to warrant publication in ACP. Here are my major concerns.

First, the authors just use the PM2.5 observations in one year, from Mar 2014 to Feb 2015, which is far from sufficient to draw any convincing conclusions. In Figure 2, they evaluate the influence of 8 different variables on PM2.5 in each season. This

means they make these conclusions using only ∼90 data values, which is far from enough. When the authors prepare this manuscript, observations in 2015 and 2016 should already be available. Why not include a longer time series of observations into this study?

Second, the discussion of the scientific significance of this work looks very superficial and unprofessional. Throughout Section 5.1, the authors made a lot of descriptive statements with little reference. For example in Line 410-413, the authors claim that rising PM2.5 concentrations prevents the occurrence of winds. Is this true? Can the authors list some references? In my understanding, the effect of aerosols on wind occurrence is much smaller than that from synoptic circulation patterns.
* * *

---

## Author Response (AR1)

Dear Dr Sally E. Pusede:

Thanks so much for giving us a chance to revise and submit our manuscript for the potential publication in ACP. According to the reviewers' comments, we managed to collect additional two years' PM2.5 and meteorological data and conduct a multiple year analysis using the CCM method. By comparing the one year's result in the previous manuscript with the multiple-year analysis result in the revised version, we found that there is no large differences at both regional and national scale, especially for the heavily polluted North China region. At the national scale, the dominant meteorological factors remain temperature, wind and humidity. The similar analysis results acquired using one year and multiple-year analysis indicated that meteorological influences on PM2.5 concentrations across China are generally stable at the inter-annual scale. In addition to the requirement for multiple-year analysis, we also added many details to the revised manuscript and the response file to explain the mechanisms of CCM. In this case, the advantage of CCM compared with the correlation analysis can be clearly demonstrated.

Reviewers suggested that the PM2.5-meteorology relationship is well-known to the chemistry society and thus we deleted this part of discussion. Instead, according to their suggestions, we added more in-depth discussion concerning the potential reasons for the large variations of meteorological influences on PM2.5 concentrations across China and potential applications of employing the p value to better estimate and predict PM2.5 concentrations.

We also addressed all the cartography and technical issues raised by reviewers.

Thanks again for considering our manuscript. We are willing to conduct any further revisions according to other requirements from you and the reviewers.

The very best

Ziyue Chen

**acp-2017-376-RC1 Anonymous Referee #3**

R: Dear Referee, thanks so much for your comments, which helped us improve this manuscript a lot. We have fully revised this manuscript according to your general and detailed comments.

Yes, as you pointed out, the principle of CCM and its advantages compared with the correlation analysis are not well-known to all scholars and thus a better explanation of the CCM method would be needed. Due to limited space, we did not add all details to the modified manuscript and some responses and explanations are given here according to your specific comments. Meanwhile, some other issues were also addressed. As your suggested that multiple year data are required, we have extended the research period from one year to three years using latest published data. Other issues are responded as follows. Thanks again for all your valuable comments and we are willing to conduct further revisions if you have further requirements.

General Comments This is an interesting paper that applies an exciting and fairly new statistical method (convergent cross mapping) to quantify the relationship between local air quality and local meteorology (wind, temperature, precipitation, etc.). The authors argue that unlike a simple correlation analysis, this method is able to demonstrate causal relationships between variables. My understanding is that this method is quite new, and it is central to this study, so I think this paper would benefit from a clearer discussion of why it is better than correlation at determining causal relationships. Would the authors' findings have differed significantly if they used correlation instead of CCM?

**R: The CCM method was proposed by Sugihara et al. (2012).**

1 Sugihara, G., May, R., Ye, H., Hsieh, C., Deyle, E., Fogarty, M., Munch, S. 2012. Detecting Causality in Complex Ecosystems. Science, 338, 496-500. Sugibara et al. (2012) pointed out that correlation analysis could extract mirage correlations, especially in complicated ecosystems. For instance, two variables A and B that have no causality may demonstrate significant correlations due to the existence of an agent variable C, which interacts with both A and B. Through a series of experiments, Sugihara et al. (2012) proved that this type of mirage correlations could be detected using the CCM method by calculating a p value of 0. The CCM method not only performed better than the correlation analysis in causality analysis by excluding the influence of other variables, but also demonstrated the advantage of detecting weak causality compared with other causality analysis method (e.g. Granger Causality), which may fail to detect weak to moderate coupling between variables.

In our previous studies, we employed both the correlation and the CCM method to examine the influence of individual meteorological factors on PM2.5 concentrations in the Beijing-Tianjin-Hebei region and compared the performance of correlation and CCM method.

Chen, Z., Cai, J., Gao, B.B., Xu, B., Dai, S., He, B., Xie, X.M. Detecting the causality influence of individual meteorological factors on local PM2.5 concentration in the Jing-Jin-Ji region. Scientific Reports 2017. 7:407352

The comparison suggests that the causality influence of individual meteorological factors on PM2.5 concentration is better revealed using the CCM method than the correlation analysis. By comparing the correlation coefficient and p value in Table 2, one can see that some correlations between meteorological factors and PM2.5 concentration may result from mirage correlations (e.g. the correlation between meanRHU and PM2.5 concentration in Hengshui in summer). Secondly, CCM analysis reveals weak or moderate coupling (e.g. the interactions between SSD and PM2.5 concentration in Cangzhou in summer) whilst correlation analysis cannot. Additionally, due to interactions between different meteorological factors, the value of correlation coefficients cannot interpret the quantitative influence of individual meteorological factors on PM2.5 concentration. Instead, the  $\rho$  value from CCM method is designed to understand the coupling between two variables by excluding influences from other factors. Through comparison, the value of the correlation coefficient for some meteorological factors is notably different from the  $\rho$  value for these meteorological factors. A large correlation coefficient for one meteorological factor may correspond to a much smaller  $\rho$  value from the CCM analysis, indicating that the value of the correlation coefficient usually overestimates the influence of individual meteorological factors on PM2.5 concentration.

The previous research (Chen et al., 2017) proved that the CCM method outperform the correlation analysis in many aspects. And this research extended the study area from the Beijing-Tianjin-Hebei region to a national scale, so the CCM method, instead of the correlation analysis, remains the ideal tool for quantifying meteorological influences on PM2.5 concentration over China.

There are some aspects of CCM that were not clear to me. Does CCM account for relationships between meteorological factors? E.g. if wind is affecting both precipitation and PM, how is the affect of wind on precipion PM counted?

R: Yes, you made a very good point here. The exclusion of influences from other variables and solely focus on the interactions between two target variables are the most important advantages of the CCM. Sugihara et al. (2012) proposed the CCM method and examined its performance in removing the influence of agent variables through a diversity of experiments. The result proved that mirage correlations caused by the influence of other variables could be detected and removed by the CCM method. For instance, Sugihara (2012) suggested that the CCM method could quantify the bi-directional interactions between two individual variables without being affected other variables, which were also proved by a diversity of studies. So for your question, for each calculation, the CCM method simply examine the bi-directional coupling between two selected variables in complicated ecosystem whilst the influence of other factors was excluded. For your 4 instance, we could calculated the coupling between PM and wind speed, and the coupling between precipitation and wind speed using the CCM method respectively, and the results for both calculations would not influence each other.

Another question I have is what produces a large value. For example, are the values higher in winter simply because there is more PM available to be effected?

R: Generally, the CCM method simply calculates the quantitative influences of individual meteorological factors on PM concentrations, whilst the mechanisms were not revealed. The mechanisms how even one individual meteorological factor influences local PM concentrations can be highly complicated. The chemical compositions, size and mass concentrations of particulate matters actually vary significantly across locations and seasons. There is no comprehensive research on how one individual meteorological factor influence PM2.5 concentrations through different mechanisms. The large  $\rho$  value in winter may result from both the much higher PM2.5 concentrations, which may be easily influenced by meteorological factors, and unique meteorological conditions. For instance, PM2.5 induced haze weather occurred frequently in winter in the Beijing, indicating a much higher PM2.5 concentration. Meanwhile, some meteorological factors in Beijing in winter were quite different from those in other seasons. For instance, the northwest wind prevails in winter and it has become a common scene and a popular saying that "the best solution for haze in Beijing is to wait for the wind". Thus the p value for wind speed in many cities within the Beijing-Tianjin-Hebei region was much larger in winter than in other seasons for both PM2.5 concentrations due to both high PM concentrations and strong wind speeds. Another instance is that the chemical composition of PM, which is related to photochemical reaction and solar radiations, also varied significantly in different seasons. Hence, the reason for the variation of p value across seasons is highly complicated, and the high PM2.5 concentration is one of major reasons. Thanks so much for pointing this out. I believe systematic research on the influencing mechanism of individual meteorological factors should be examined in-depth by scholars from a diversity of backgrounds.

Or as another example, is precipitation more effective at removing PM along the coasts because it rains more? If there were a way to normalize by the amount of total rainfall, would precip still be more important along the coasts than in the drier interior?

R: Yes, it is highly possible that precipitation is more effective at removing PM along the coasts because it rains at a higher frequency and intensity. As we know that the PM2.5 concentrations drop significantly after a heavy rain whilst light rain may not reduce PM2.5 concentrations significantly. Meanwhile, PM2.5 concentrations may also affect the influence of precipitation. Light rains may have limited washing-off effects on high-concentration PM2.5 concentrations and may increase the relative humidity in the environment, which is favorable for the rising of PM2.5 concentrations. In the drier interior, the PM2.5 concentrations are usually much higher and the intensity and frequency of precipitation are much lower than those along coasts. These two factors may both be the reason that precipitation is more effective at removing PM along the coasts. Due to these influencing factors, precipitation may still be a less important meteorological driving force for PM2.5 concentrations in the drier interior, even if there were a way to normalize by the amount of total rainfall. In the revised manuscript, we added a new part to briefly introduce the underlying reasons for the variation of

My second major concern is that this study uses a single year of data to make general comments about PM-meteorology relationships. This gives us little sense of how much year-to-year variability may exist in these relationships and generally weakens the conclusions. If CCM is too computationally expensive to use on multiple years, perhaps a different method could be used to supplement it. R: This is a very good point. When we were preparing this manuscript, there were only one-year's data available. And now, we endeavored to acquire another two years' data for multiple year analysis. Although the CCM is computationally expensive, we still believe it is more persuasive to use multiple years' data for drawing conclusions. So thanks so much for this suggestions. We have re-calculated the CCM results using three years' data, and the updated results have been added to the revised manuscript. By comparing the mean annual p value for eight variables for 2014 with the mean annual p value for eight variables for 2014 with the mean annual p value for eight variables during 2014-2016, we can see that the meteorological influences on PM2.5 concentrations at the national scale did not change significantly in the past three years.

Finally, I would like to see a deeper discussion of the scientific significance of this work. As it is written currently, this paper is almost purely descriptive. The new method is interesting, but the paper could do a better job of articulating what we are learning from it. Perhaps some discussion about why different meteorological factors are more/less important in different regions/seasons, for example, would help give the paper more depth. I would also suggest spending more time discussing the implications for modeling, especially in the introduction and conclusions, as that was what I took away as the most important implication in this paper.

R: Thanks so much for this valuable suggestion. Yes, it is important to provide more indepth discussion for the acquired results based on the CCM. And as you suggested, some discussion, e.g. the mechanisms between meteorology factors and PM2.5 concentrations were well-known, and it is not necessary to introduce it in details. So in the revised manuscript, we have fully removed this part and left more space for the parts you suggested. We added some introduction on the potential applications the CCM method, instead of the correlation analysis, in research concerning meteorology-included PM2.5 concentration estimation prediction. Meanwhile, the potential reason for large variations of meteorological influences on PM2.5 concentrations across China has been added to the revised version. Thanks again for this valuable comment.

**Specific Comments**

pp 3, ln 93-95 - Can you elaborate further on how your previous study showed that CCM is better than correlation for the benefit of readers who have not read that paper. It is important for your results here to make as clear as possible why CCM is a better method/provides new insights.

R: This is a very good point. We have mentioned the advantages of the CCM method, compared with the correlation analysis in the method parts. However, as you pointed out, we should give more information on how the CCM method performed better than the correlation analysis. So we added some extra details concerning the CCM method in the method part (since Line 197).

As mentioned above, our previous study (Chen et al., 2017) employed the correlation analysis and the CCM method to examine the influence of individual meteorological factors on PM2.5 concentrations in the Beijing-Tianjin-Hebei region. In the paper, we demonstrated detailed results of both correlation and causality influence in two large tables. By comparing these results, we found that (1) some correlations between meteorological factors and PM2.5 concentration may result from mirage correlations. For instance, there was a correlation between meanRHU and PM2.5 concentration in Hengshui in summer whilst the causality influence of meanRHU on PM2.5 concentrations is 0, indicating a mirage correlation.

(2) CCM analysis reveals weak or moderate coupling whilst correlation analysis cannot. For instance, the correlation between SSD and PM2.5 concentration in Cangzhou in summer was not significant whilst there is a weak causality influence of SSD on PM2.5 concentrations detected. (3) Due to interactions between different meteorological factors, the value of correlation coefficients cannot interpret the quantitative influence of individual meteorological factors on PM2.5 concentration. Instead, the  $\rho$  value from CCM method is designed to understand the coupling between two variables by excluding influences from other factors. Through comparison, the value of the correlation coefficient for some meteorological factors is notably different from the  $\rho$  value for these meteorological factors. A large correlation coefficient for one meteorological factor may correspond to a much smaller  $\rho$  value from the CCM analysis. We found that the correlation coefficient between individual meteorological factors and PM2.5 concentrations was usually much larger than the value. This indicated that the causality of individual meteorological factors on PM2.5 concentrations was generally overestimated using the correlation analysis, due to the influences from other meteorological factors.

So the  $\rho$  value is a more reliable indicator for understanding and comparing quantitative influences of different individual meteorological factors than correlation coefficient.

Due to limited spaces, in the previous manuscript, we did not give all information on the comparison between correlation analysis and the CCM method. But according to your suggestion, more details have been added to the revised manuscript

**pp 8 ln 221 - Are the results sensitive to the choice of parameters?**

R: The CCM detects the bi-directional coupling between two variables highly automatically and only several parameters are required to run this model. However, the CCM result mainly depends on the time series data of the two variables and the several parameters mainly influence the presentation effects of the Convergent maps. We have tested different setting of these parameters, and the extracted  $\rho$  value was simply the same, only the presentation of convergent maps were more smooth with different settings. So the results were not sensitive to the choice of parameters, which is also the major advantage of the CCM method. Since CCM method is not sensitive to the choice of parameters and the reliability of the causality analysis result has been proved by hundreds of studies from different ecosystems, this method has been widely employed.

pp 9 ln 237 – how is the value of determined? Are you calculating the limit or taking the value at a specific time series length?

**R**: This is a good question. When we determine whether a curve is convergent or not, we set a  $\Delta$  to represent the variation of  $\rho$  value along with increasing time series length. If the  $\Delta$  was less than a given threshold (e.g. 0.01) for a consecutive several date until the end of the time series, we consider that the  $\rho$  was convergent to the p value of last date.

What about cases such as PM2.5 xmap minTEM, which (at least by eye) does not appear to be converging? Wouldn't that suggest that minTEM was not influencing PM2.5 at all?

R: The CCM method considers strict causality influence of one variable on the other one and if the curve is convergent to 0 or demonstrates no generally convergent trend, then no causality influence exists based on long-term time series analysis of two variables. And for your instance, PM2.5 xmap minTEM demonstrate a generally convergent trend to a value approximating to 0.2. You may argue that this curve is not strictly convergent. However, the CCM method does not actually conducts a limit calculation and the CCM curve simply demonstrates a general trend of convergence. According to a diversity of instances provided by Sugihara et al. (2012), the curve shape of PM2.5 xmap minTEM can be regarded as a convergent curve, indicating a detected causality. Those instances from Sugihara et al. (2012) that demonstrated a non-convergent trend are near-linear shape or totally irregular shapes, which are quite different from the PM2.5 xmap minTEM. In summary, according to the PM2.5 xmap minTEM, minTEM exerted a weak influences on PM2.5 concentrations.

pp 9 ln 244 - It's not clear here how PM is changing wind speed

R: High PM2.5 concentrations may lead to haze episodes, which usually result in generally stable atmospheric environment. In this case, the formation of winds, especially strong 10

winds within this atmospheric environment are influenced significantly. And this is the reason why there are few winds within the urban areas during severe haze episodes. Meanwhile, winds across regions with haze episodes are also influenced notably.

Yang et al. (2015) observed four haze episodes during Oct to Nov, 2014 and during these four haze episodes in the North China plain, the very high PM2.5 concentrations all led to stagnant condition and weak high-pressure systems, which further led to slowed wind speed and disturbed wind direction. This phenomenon was also observed by Liu et al. (2014) in haze episodes in Beijing in 2013. Very high PM2.5 concentrations induced haze episodes further led to stagnant and stable high-pressure systems, which made megacities served as obstacles to significantly slowed down the wind speed (Yang et al., 2015). Therefore, the effects of aerosols, especially high-concentration PM2.5 concentrations, prevented the wind occurrence mainly through indirect mechanisms.

Yang, Y. R., Liu, X. G., Qu, Y., An, J. L., Jiang, R., & Zhang, Y. H., et al. (2015). Characteristics and formation mechanism of continuous hazes in china: a case study during the autumn of 2014 in the north china plain. Atmospheric Chemistry & Physics, 15(14), 10987-11029.

Liu, X. G., Li, J., Qu, Y., Han, T., Hou, L., & Gu, J., et al. (2013). Formation and evolution mechanism of regional haze: a case study in the megacity beijing, china. Atmospheric Chemistry & Physics, 13(9), 4501-4514.

pp 10 ln 270 - I would guess that (out of the 189 cities) those clustered regionally would show similar maps. Is that true? I.e. are the 37 cities shown representative of the cities not shown?

R: The main reason we selected some representative cities, instead of all cities for presenting wind rose maps is that there is limited space in the map. As one can see, it is already very crowded to demonstrate 37 wind roses and we should pay special attention to the selection of representative cities. Since the provincial capital in each province is the most important city( and usually the largest city) within the province and thus the meteorological influences on  $PM_{2.5}$  concentrations for the provincial capital city usually receive the most emphasis. In this case, we selected the provincial capital for 31 provinces to present meteorological influences on  $PM_{2.5}$  concentration in each province, which considers both the spatial locations and the importance of the representative cities. For most provinces, especially provinces with low  $PM_{2.5}$  concentrations, the number of monitoring cities and the variations of  $PM_{2.5}$  concentrations are small. For regions with heavy air pollution (e,g, , the number of monitoring cities and the variations of  $PM_{2.5}$  concentrations are small. In that case, we believe that the selective cities can be representative of no-shown cities within the same province.

pp 13 ln 292-296 - This only seems to be true in some seasons.

**R**: Thanks so much for this point. We also noticed that this part should be described more rigorously. So in the revised manuscript, we have fully revised this part.

pp 13 ln 299-300 - Can you quantify this more rigorously? By eye, it seems like there are enough outliers to call this in to question

R: Thanks for pointing this out. We realized that this conclusion may not apply to all regions in China. Instead, this pattern " the higher PM2.5 concentrations, the stronger meteorological influences on PM2.5 concentrations" was most obvious for the North China region, which is the most polluted region in China. In the revised manuscript, we have rephrased this statement to "For the North China region". Thanks again for this suggestions.

pp 14 ln 337-339 - This seems true for winter vs. summer, but what about spring vs autumn?

R: Again, thanks for pointing this out. Generally,  $PM_{2.5}$  concentrations for one specific city are highest in winter and the lowest in summer. So in the paper, when we mention the season when  $PM_{2.5}$  concentration is high (low), we mainly mean winter (summer). Meanwhile, the characteristics of  $PM_{2.5}$  concentrations in spring and autumn are not obvious, so we mainly compare the characteristics of  $PM_{2.5}$  concentrations in winter and summer, which provides the most important reference for the management of  $PM_{2.5}$ induced haze episodes in winter. So thanks again for this comments and we realized that we should make the statement clearer to avoid some confusion. In the revised manuscript, this sentence has been rephrased.

pp 18 Fig 4 caption – Is there a particular argument for only including the dominant factor in each city?

R: Thanks for this comment. As explained in the text, there are more than 189 cities for this research and it requires some space in the map to place the wind rose map for each city without overlapping with each other. And as you can see, the wind rose map for 37 representative cities has already been filled with 37 wind roses with different sizes and it is impossible to present 189 wind roses on a single map without severe overlapping effects. So we employed an alternative approach to only present the dominant meteorological factor for all cities and an entire wind rose for 37 representative cities in separate maps.

Section 5.1 of the discussion feels out of place. This is more of a discussion of what we already know about aerosol-meteorology interactions than a discussion of the implications of the work done in this study. I would recommend either rewriting it so that it builds more on the results from this paper, or cut it and integrate the important information into earlier sections.

**R**: Thanks so much for your comments. As also suggested by another referee, this part has been well known to scholars with relevant background. So in the revised manuscript,

**this part has been totally cut. Meanwhile, we have added a separate paragraph to explain the underlying reasons for large variations of meteorological influences**

pp 24 ln 562-566 – as per my earlier comment, I am not sure that you have shown this. If Beijing were to receive the same amount of precipitation that a coastal city does, is it possible that precipitation would become more important in Beijing? Does looking at how a specific factor changed PM in a year tell us about how effective changes in that factor would be?

**R**: Thanks for pointing this out. I thought it is a bit difficult to test the hypothesis using the CCM method. This is because the precipitation amount for most days in Beijing is 0 and thus it is difficult to normalize the 0 value to a value similar to that in coastal cities. As an alternative solution, as responded above, we added another two years data to conduct a multiple-year analysis and checked whether the influence of precipitation on PM2.5 concentrations in Beijing vary with longer time series.

Technical Corrections pp 1, ln 17 (and later occurrences) - "causality influence" is redundant. "Influence" already implies causality.

**R: All Corrected.**

pp 2, ln 35-36 - "Amongst these environmental elements, . . . concerned social and ecological issues." The wording of this sentence is unclear.

**R: This sentence has been rephrased.**

pp 2, ln 42 - "Serious haze not only influences peoples daily life," this wording is vague. How does haze influence peoples daily lives?

R: Serious haze episodes, usually presented as very thick and heavy black fogs, cause serious negative influences on people's daily life, especially the traffic. During a severe 14

haze episode in Beijing in January, the extreme high fog episodes led to very low visibility and heavy traffic jam. This is one instance how haze influence people's daily life.

pp 2, ln 57 - "controversy" should be changed to "controversial"

**R:** Corrected.**

pp 3, ln 68 - "... fractions of three different sizes. .." this is unclear. Authors should indicate that they are talking about aerosols, and specify the sizes.

**R: Thanks for pointing this out. We have corrected this in the revised manuscript.**

pp 3, ln 7 - what region is this study referring to?

**R: The missing "research region" content has been added to the revised manuscript.**

pp 3, ln 86 - word choice: I would suggest "well studied" instead of "massively studied"

**R:** Corrected.**

pp 4, ln 119 - what does API stand for?

**R: API (Application Programming Interface).**

pp 5, ln 136-137 - I'm not clear on what small and large evaporation are.

R: Small evaporation indicates the evaporation amount calculated using a small-diameter measurement equipment and large evaporation indicates the evaporation amount calculated using a large-diameter measurement equipment. Generally, the amount calculated using the two types of equipment is usually the same, although slightly difference may exist between measured evaporation values. And the p value for the factor evaporation is decided by the larger value of the two indicators.

pp 5, ln 144 – sunshine duration for the day is a less widely used term and should be defined here

R: Thanks so much for pointing this out. Sunshine duration or sunshine hours is a climatological indicator, measuring duration of sunshine in given period (usually, a day or a year) for a given location on Earth. This definition has been added to the modified manuscript.

pp 5 ln 146 - what qualifies as extreme wind speed?

R: The max wind speed indicates the max mean wind speed during any 10 minutes within a day's time.

The extreme wind speed indicates the max instant (for 1s) wind speed within a day's time.

Thanks for pointing this out and the definition has been added to the modified manuscript.

pp 5 ln 146 - how is max wind direction defined?

R: The max wind direction indicates the dominant wind direction for the period with the max wind speed.

pp 6 ln 174 – "Two time series" is unnecessary. Suggest changing the sentence to " $\{X\}=[X(1), \ldots X(L)]$  and  $\{Y\}=[Y(1), \ldots Y(L)]$  are defined as the temporal variations of variables X and Y."

**R: Corrected according to your suggestions.**

pp 6 ln 175 – It's unclear what r and S are.

**R:** Thanks for pointing this out.

r is the current position in the time series and the S presents the start position in the time series.

pp 8 ln 217 – why can E be 2 or 3?

R: This parameter decides at what dimension the CCM was calculated. When E equals 3, the calculation accuracy is higher. Through experiments, we found that the results were generally

the same using the value of 2 or 3. For this research, we set the value of E 3 for a theoretically optimal CCM result.

pp 14 ln 331-332 - the phrasing here is unclear.

**R: This sentence has been rephrased.**

pp 18 ln 374-376 - is there a way to test if these values are significant?

R: The CCM method is different from the correlation analysis and another classic causality analysis method , Granger causality, which provides readers with the significance for the coupling between two variables. The CCM did not give us a value to present the significance for the revealed causality. However, while the Granger causality mainly revealed the qualitative causality between two variables, the p value from the CCM method revealed the quantitative causality between them. And the CCM method suggests that if the significance between two variables was not significant, then the calculated p value would be 0. So the p value was a direct metric for the quantitative influence and an indirect metric for the significance.

pp 20 ln 419 - this wording is unclear

**R: This part has been removed in the revised manuscript.**

pp 20 ln 426 – Wikipedia is not an appropriate source. Better to cite a scientific paper that defines SSD.

R: In the revised manuscript, the PM2.5-meteorology interaction part has been removed according to your and other referees' comments. However, here we would like specifically add the explanation here that SSD could reduce PM2.5 concentrations not only through

atmospheric photolysis, but also by enhancing surface temperature and promoting upward movement of aerosols.

pp 20 ln 440 – Temperature inversion is certainly important, but none of the metrics in this study measure it directly.

R: Thanks for pointing this out. Although this part has been removed in the revised manuscript, we would like to add some other mechanisms. Actually, in addition to temperature inversion, another important mechanism is that Temperature is closely related to pollutant concentrations by affecting atmospheric turbulence and chemical reactions. The temperature is positively correlated with pollutant concentrations in the majority of cities (He et al., 2017).

He, J., Gong, S., Ye, Y., Yu, L., Lin, W., Mao, H., et al. (2017). Air pollution characteristics and their relation to meteorological conditions during 2014–2015 in major chinese cities. Environmental Pollution, 223, 484-496.

pp 21 ln 447 - what about horizontal transport (advection)?

R: Yes, horizontal transport of airborne pollutants should be major reason for the variation of PM2.5 concentrations. i.e. Anticyclones (i.e., high pressure systems) induced low wind speed was not favorable for the dispersion of pollutants. On the other hand, low pressure systems may lead to large wind speeds, favorable for the dispersion of PM2.5.

pp 21 ln 446 – change "social economic" to "socio-economic" pp 22 ln 492 – change "negative causality on" to "decreases" and "positive causality on" to "increases"

**R:** Corrected.**

pp 23 ln 527-528 - do you have citations for this?

**R: We have added more relevant references to the revised manuscript according to your**

comments.

pp 24 ln 544 – can you give more details/citations about the controversy?

R: Some reports concerning different effects of this project has been added to the revised manuscript.

**acp-2017-376-RC2 Anonymous Referee #1**
Thanks so much for your encouragement and valuable comments. We have fully revised this manuscript according to your general and detailed comments, as well as comments from other reviewers. We would like to make further revision in due stages if you have further requirements.

Page 2, lines 56-57: "Although quantitative contributions of different sources (e.g. coal burning and automobile exhaust) to airborne pollutants remain controversy" – It's not clear what you mean here with the "controversial" – politically or scientifically? If scientifically – the direct emissions and/or subsequent chemistry?

R: Yes, the controversy mainly comes from the mixed understanding of relative contribution. For instance, some scholars claimed that automobile exhaust took up only 4% of relative contributions to PM2.5 concentrations. However, many following papers

argued that the actual contribution from automobile exhaust took up more than 20%. The difference was that the former mainly considered the direct emissions whilst the latter ones comprehensively considered the direct emission and following secondary pollutants. So yes, your point was exactly the situation.

Page 4, section 2.1.1: How do you quality control the data and/or deal with missing data?

R: For this research, all released data were previously maintained by specific institutions and there are several stations for each city to report hourly PM2.5 concentrations conditions. For some stations, missing data lead to 0 value. If there are stations with Non-0 value, then the mean PM2.5 concentration for a specific city was calculated using these stations. So for most days in each city, a valid mean PM2.5 concentration value could be calculated. For days when the measured PM2.5 concentration from all stations was 0, then mean PM2.5 concentration was 0. The record for this day was deleted. Only a very small proportion of cities experienced days with no daily average data. And since few missing records would not influence the order of time series of PM2.5 and meteorological data, the CCM result would not be influenced by the missing data. Meanwhile, for cities (e.g. Liaocheng and Zhuji) with a large amount of missing meteorological data, we deleted this city for this research.

CCM method: How does the time lag parameter affect the results? The resolution of the map is mentioned but how does it affect the physical interpretation of the results? – Especially for those variables that may act on a shorter time scale.

R: We compared the CCM analysis result calculated using different parameters: 2,5, 10, 20 and the result was generally the same. Just the resolution of the map was higher with a small time lag. And you made a good point here that the physical interpretation of the results may lead to biased p value. Actually, the presented CCM map was simply for a 21

basic demonstration about how CCM works. For exact p value, the provided CCM algorithm actually calculated an accurate p value with the increase of time series, and the CCM map was produced based on a series of accurate p values. So for this research, it is not feasible and reliable to physically interpret the p value for 190\*18\*4 CCM maps and the p value used for producing the wind-rose and other maps were extracted directly from the program.

Page 13, lines 295-296: This causality seems to be backwards: i.e., why would differences in PM levels cause differences in meteorological influences? What mechanism would cause this?

R: This is a very good question. Actually, this phenomenon was revealed and proposed in Chen et al. (2017). Chen et al. (2017) found that in the Beijing-Tianjin-Hebei region, the causality influence of individual meteorological factors on PM2.5 concentrations was the strongest in winter, when the PM2.5 concentrations were the highest, for all cities, Meanwhile, the causality influence of individual meteorological factors on PM2.5 concentrations was the weakest in winter, when the PM2.5 concentrations were the lowest. The potential mechanism could be that similar meteorological conditions may lead to large variations of PM2.5 concentrations were high and may lead to haze episodes, a strong northwester wind may immediately reduce the PM2.5 concentrations to a very low level. Meanwhile, high wind-speed in summer may lead to small variations of PM2.5 concentrations, as the original PM2.5 concentrations are low. Similarly, other meteorological factors are more likely to change PM2.5 concentrations significantly when the PM2.5 concentrations are high. In the revised manuscript, we have rephrased this part to avoid unnecessary confusions.

Page 14, lines 330-333: This sentence is very vague - can you be more specific?

R: Thanks so much for pointing this out. Yes, this part should be explained with more details. Actually, what we mean here is that some meteorological factors can be dominant factors across China. For instance, according to Fig 3, you can see such factors as temperature and wind were dominant meteorological factors in many regions, including Northeast, Northwest, coastal areas and inland areas; Meanwhile, some meteorological factors for limited regions (Mainly middle inland cities). This part has been added to the revised manuscript.

Figure 2 and 3: I would suggest the background of concentrations be in a gray scale so the colored icons/wind roses stand out more. Also, how different would the maps be if the correlation coefficient were used instead? A statement or two would reinforce the argument for the use of CCM rather than correlation coefficient.

R: Thanks so much for the cartography suggestions. We have updated the Fig 2 and 3 according to your suggestions and made some further revisions to improve the quality of maps.

Another referee also mentioned that the difference between the p value and the correlation coefficient. And we are sorry that we did not make this clear in previous manuscript. Here we simply explained the advantages of CCM method and some findings concerning the comparison between correlation analysis and the CCM method from our previous studies.

The CCM method was proposed by Sugihara et al. (2012).

1 Sugihara, G., May, R., Ye, H., Hsieh, C., Deyle, E., Fogarty, M., Munch, S. 2012. Detecting Causality in Complex Ecosystems. Science, 338, 496-500.

Sugibara et al. (2012) pointed out that correlation analysis could extract mirage correlations, especially in complicated ecosystems. For instance, two variables A and B that have no causality may demonstrate significant correlations due to the existence of an agent variable C, which interacts with both A and B. Through a series of experiments, Sugihara et al. (2012) proved that this type of mirage correlations could be detected using the CCM method by calculating a p value of 0. The CCM method not only performed better than the correlation analysis in causality analysis by excluding the influence of other variables, but also demonstrated the advantage of detecting weak causality compared with other causality analysis method (e.g. Granger Causality), which may fail to detect weak to moderate coupling between variables.

In our previous studies, we employed both the correlation and the CCM method to examine the influence of individual meteorological factors on PM2.5 concentrations in the Beijing-Tianjin-Hebei region and compared the performance of correlation and CCM method.

Chen, Z., Cai, J., Gao, B.B., Xu, B., Dai, S., He, B., Xie, X.M. Detecting the causality influence of individual meteorological factors on local PM2.5 concentration in the Jing-Jin-Ji region. Scientific Reports 2017. 7:407352

The comparison suggests that the causality influence of individual meteorological factors on PM2.5 concentration is better revealed using the CCM method than the correlation analysis. By comparing the correlation coefficient and  $\rho$  value in Table 2, one can see that some correlations between meteorological factors and PM2.5 concentration may result from mirage correlations (e.g. the correlation between meanRHU and PM2.5 concentration in Hengshui in summer). Secondly, CCM analysis reveals weak or moderate coupling (e.g. the interactions between SSD and PM2.5 concentration in Cangzhou in summer) whilst correlation analysis cannot. Additionally, due to interactions 24 between different meteorological factors, the value of correlation coefficients cannot interpret the quantitative influence of individual meteorological factors on PM2.5 concentration. Instead, the  $\rho$  value from CCM method is designed to understand the coupling between two variables by excluding influences from other factors. Through comparison, the value of the correlation coefficient for some meteorological factors is notably different from the  $\rho$  value for these meteorological factors. A large correlation coefficient for one meteorological factor may correspond to a much smaller  $\rho$  value from the CCM analysis.

The previous research (Chen et al., 2017) proved that the CCM method outperform the correlation analysis in many aspects.

Page 20, lines 414-420: While higher relative humidity does lead to hygroscopic growth of aerosols, this is probably not evident in the observed concentrations since most measurements are taken at a constant relative humidity (e.g, 35% in US and Europe). Measurements in China may not do this, and if so, should be explicitly stated since this can have a major effect to aerosol mass depending on the composition of the aerosol.

R: Thanks so much for this explanation. This information is very useful for future comparison of meteorological influences, especially the humidity factor, on PM2.5 concentrations in China and other regions. The reason we added the general introduction of mechanisms how meteorological factors may interact with PM2.5 concentration is that one referee during the first stage of ACPD review process suggested we do so. However, during this round of ACPD review, you and other referee all suggested that the part of introduction is well known to scholars with meteorological background and we have deleted this part in the revised manuscript.

Page 23, lines 515-535: This paragraph seems out of place with the rest of the section. Page 23, line 525: I am not able to read Cheng et al. (2015), but I'm wondering what the model is using for predictors? If they are "static" models, isn't that just the mean state? I'm having a hard time understanding. If the argument is to use CCM instead of correlations, an example (see above) would help to reinforce this.

R: Static statistical models did not consider the influence of meteorology on PM2.5 concentrations whilst dynamic models select some reliable and key meteorological influencing factors for better predicting PM2.5 concentrations. The advantage of p value compared with the correlation analysis has been explained above and added to the revised manuscript. As suggested by another referee, the improvement of models based on the CCM method could be important practical applications of the meteorological influences on PM2.5. So more in-depth discussion concerning this part has been added to the revised manuscript.

Page 24, lines 562-566: How does the frequency of precipitation affect this statement? For example, if precipitation is rare in Beijing during winter, especially compared to the Yangtze River Basin.

R: This is a very good point and has also been pointed out by another referee. Yes, it is highly possible that precipitation is more effective at removing PM along the coasts because it rains at a higher frequency and intensity. As we know that the  $PM_{2.5}$ concentrations drop significantly after a heavy rain whilst light rain may not reduce  $PM_{2.5}$ concentrations significantly. Meanwhile,  $PM_{2.5}$  concentrations may also affect the influence of precipitation. Light rains may have limited washing-off effects on highconcentration  $PM_{2.5}$  concentrations and may increase the relative humidity in the 26 environment, which is favorable for the rising of  $PM_{2.5}$  concentrations. In the drier interior, the  $PM_{2.5}$  concentrations are usually much higher and the intensity and frequency of precipitation are much lower than those along coasts. These two factors may both be the reason that precipitation is more effective at removing PM along the coasts. Due to these influencing factors, precipitation may still be a less important meteorological driving force for  $PM_{2.5}$  concentrations in the drier interior, even if there were a way to normalize by the amount of total rainfall. We have added more discuss on this in the revised manuscript.

Results/Discussion: Much of this review of meteorology-PM2.5 relationships in the discussion would probably be better suited in the introduction and in the results as it pertains to different locations within China. Many of the statements in the results are rather vague (e.g., page 14, line 330-333) and could be elaborated to include specific meteorological factors and specific locations.

R: Thanks so much for your comments. As explained above, the review of meteorology-PM2.5 relationships has been removed in the revised manuscript. And some vague statements in the previous manuscript have been re-phrased with more details.

Minor comments

Page 2, line 61: were correlated

R: Corrected

Page 3, line 68: "fractions of three different sizes" of particulate matter

**R: Corrected**

Page 4, lines 119-120: What does this sentence mean?

**R**: Sorry that we did not make this clear. The API (Application Programming Interface) tool we programmed can automatically downloaded hourly air pollution data since the execution of this tool.

Page 12, line 288: Awkward wording

**R: Rephrased**

Page 20, line 426: Wikipedia is not an appropriate citation.

R: Other definition has been added in the revised manuscript in other parts.

Page 20, line 427 and elsewhere: Check your usage of "by analogy" – you may be looking for a different phrase.

R: Thanks so much for this suggestion. We have changed this to "similarly".

Page 20, line 433 and elsewhere: Check subject verb agreement, specifically for "PM2.5" and "concentration(s)"

R: Thanks so much for this. We have corrected all these incorrectly used format.

**acp-2017-376-RC3 Anonymous Referee #2**

**Dear Referee:**

Thanks so much for your valuable suggestions. We have fully revised the manuscript according to your suggestions in the revised manuscript. And we are willing to conduct further revision if you have additional requests.

This paper attempts to investigate the meteorological influence on PM2.5 concentra- tions in China at the national scale using a convergent cross-mapping (CCM) method. This method is somewhat new to the atmospheric chemistry community, but the physi- cal mechanism as discussed in this paper is very descriptive and already well-known. Overall I don't feel these results are significant enough to warrant publication in ACP. Here are my major concerns.

R: Thanks so much for pointing this out. Actually, the major aim of this research is to quantify the causality influence of individual meteorological factors on  $PM_{2.5}$  concentrations in 190 monitoring cities across China. The spatial and seasonal variations of meteorological influences on  $PM_{2.5}$  concentrations at a national scale have rarely been examined before. Meanwhile, previous studies for meteorological influences on  $PM_{2.5}$  concentrations at local and regional scale mainly employed the Correlation analysis, which can lead to mirage correlations and unreliable correlation coefficient, due to complicated interactions between different meteorological factors. Thus the use of CCM method has the advantage to remove potential influences from other variables when analyzing the bi-directional coupling between two variables. The comparison and patterns of calculated p value (quantitative causality influence) of individual meteorological factors on  $PM_{2.5}$  concentrations across China is the key findings for this research.

Yes, as you pointed out, the e physical mechanism We did not add the physical mechanism of  $PM_{2.5}$ -meteorology relationship in the first version of manuscript. One reviewer in the 29

first stage of ACPD review suggested this, and thus we added a brief discussion. However, in this round review, you and other referees all pointed out that this part was off the structure and was already well known to scholars with relevant background. So in the revised manuscript, according to your suggestions, we have deleted this part. And for other major issues you pointed out, e.g. the lack of multiple year analysis ,we have fully revised this manuscript accordingly and explained as follows. Thanks again for your valuable comments and we would like to make further revision in due stages if you have further requirements.

First, the authors just use the PM2.5 observations in one year, from Mar 2014 to Feb 2015, which is far from sufficient to draw any convincing conclusions. In Figure 2, they evaluate the influence of 8 different variables on PM2.5 in each season. This means they make these conclusions using only 90 data values, which is far from enough. When the authors prepare this manuscript, observations in 2015 and 2016 should already be available. Why not include a longer time series of observations into this study?

R: This is a very good point. Long-term observation data are more likely to present reliable causality influence of 8 different variables on PM2.5 in each season, as one-year data may be influenced by abnormal meteorological conditions. So according to your suggestions, we managed to collect the PM2.5 and meteorological data from Mar 2014 to Feb 2017. In the revised manuscript, we have added additional two years' data for multiple-year analysis and thus a comprehensive CCM analysis based on three year's analysis has been conducted. Thanks again for pointing this out, as the inclusion of multiple-year analysis made the results more robust.

Second, the discussion of the scientific significance of this work looks very superficial and unprofessional. Throughout Section 5.1, the authors made a lot of descriptive statements with

little reference. For example in Line 410-413, the authors claim that rising PM2.5 concentrations prevents the occurrence of winds. Is this true? Can the authors list some references? In my understanding, the effect of aerosols on wind occurrence is much smaller than that from synoptic circulation patterns.

**R**: Thanks so much for this valuable suggestions. As explained above, we also know that  $PM_{2.5}$ —meteorology interactions, as you and another two referees pointed out, the mechanisms were well-known and may be off the focus of this manuscript. For this reason, we did not add this introduction of this part to the original manuscript. During the first stage of ACPD discussion, a referee kindly suggested that a brief introduction of PM2.5-meteorology relationship can be added, and thus we provided a general introduction of mechanisms in the previous manuscript. In the revised manuscript, according to the suggestions of you and other referees, we have deleted this part to make the aim and key findings highlighted. In addition, according to the comments of you and other referees, we have added some more in-depth discussion, concerning the potential applications of this research and underlying reasons for the large variations of meteorological influences on  $PM_{2.5}$  concentrations across China, has been added to the revised manuscript.

Although the PM2.5—meteorology interaction part has been removed, we would like to give some explanations on the example you suggested. Yes, we understood that synoptic circulation patterns were the major causes for wind occurrence and we are not claiming that the effects of aerosols were large enough compared with the synoptic circulations. We just pointed out that the potential mechanisms of the negative feedbacks of high PM2.5 concentrations

Yang et al. (2015) observed four haze episodes during Oct to Nov, 2014 and during these four haze episodes in the North China plain, the very high PM2.5 concentrations all led to

stagnant condition and weak high-pressure systems, which further led to slowed wind speed and disturbed wind direction. This phenomenon was also observed by Liu et al. (2014) in haze episodes in Beijing in 2013. Very high PM2.5 concentrations induced haze episodes further led to stagnant and stable high-pressure systems, which made megacities serve as obstacles to significantly slow down the wind speed (Yang et al., 2015). Therefore, the effects of aerosols, especially high-concentration PM2.5 concentrations, prevented the wind occurrence mainly through indirect mechanisms.

Yang, Y. R., Liu, X. G., Qu, Y., An, J. L., Jiang, R., & Zhang, Y. H., et al. (2015). Characteristics and formation mechanism of continuous hazes in china: a case study during the autumn of 2014 in the north china plain. Atmospheric Chemistry & Physics, 15(14), 10987-11029.

Liu, X. G., Li, J., Qu, Y., Han, T., Hou, L., & Gu, J., et al. (2013). Formation and evolution mechanism of regional haze: a case study in the megacity beijing, china. Atmospheric Chemistry & Physics, 13(9), 4501-45

**Understanding meteorological influences on PM2.5 concentrations across China:**

`a temporal and spatial perspective

Ziyue Chen1,2, Xiaoming Xie1, Jun Cai3, Danlu Chen1, Bingbo Gao4, Bin He1,2,

Nianliang Cheng5, Bing Xu3\*

[revised manuscript text omitted]
  $\mathbf{x}(t) = \langle X(t), X(t-\tau), \dots, \tau \rangle$ ,  $X(t-(E-1)\tau \tau) >$  for  $t = 1+(E-1)\tau \tau$  to t = r. To generate
- a cross-mapped estimate of  $Y(t) \leftarrow \hat{Y}(\hat{Y}(t)|M_X)$ , the contemporaneous lagged-coordinate

|-------------------------|
|                         |

| 154 | vector on $M_x$ , x(t) is located, and then its E+1 nearest neighbors are extracted, where E+1                  |
|-----|-----------------------------------------------------------------------------------------------------------------|
| 155 | is the minimum number of points required for a bounding simplex in an E-dimensional                             |
| 156 | space (Sugihara and May, 1990). Next, the time index of the $E+1$ nearest neighbors of $x(t)$                   |
| 157 | is denoted as $t_1$ ,, $t_{E+1}$ . These time index are used to identify neighbor points in $\underline{Y}$ and |
| 158 | then estimate $Y(t)$ according to a locally weighted mean of E+1 $Y(t_i)$ values (Equation 1).                  |

159

$$\hat{Y}(t)|M_x = \sum_{i=1}^{k+1} w_i Y(t_i)$$
(E1)

Where  $w_i$  is a weight calculated according to the distance between X(t) and its ith nearest neighbor on  $M_X$ .  $Y(t_i)$  are contemporaneous values of Y. The weight  $w_i$  is determined according to Equation

162

2.

| 163 | $w_i = u_i \Big/ \sum_{j=1}^{E+1} u_j$ | (E2) |
|-----|----------------------------------------|------|
|     |                                        |      |

164 165 Where  $u_i = e^{-d[\underline{x}(t),\underline{x}(t_i)]/d[\underline{x}(t),\underline{x}(t_i)]}$  whilst  $d[\underline{x}(t), \underline{x}(t_i)]$  represents the Euclidean distance between two vectors.

166 In our previous research, interactions between the air quality in neighboring cities (Chen, 167 Z. et al., 2016), and bidirectional coupling between individual meteorological factors and 168 PM2.5 concentrations (Chen, Z. et al., 2017) were quantified effectively using the CCM 169 method. By comparing the performance of correlation analysis and CCM method, Chen, 170 Z-et al. (2017) proved that the CCM method not only detected mirage correlations, but 171 also extracted weak coupling, which may not be detected by correlation analysis. Additionally, Chen, Z et al. (2017) indicated that the -P-value was a more reliable 172 173 indicator of quantitative meteorological influences on PM2.5 concentrations than the 174 correlation coefficient. et al. (2017) suggested that correlation analysis may lead to a 175 diversity of biases due to complicated interactions between individual meteorological 176 factors. Firstly, some mirage correlations (two variables with a moderate correlation 177 coefficient) extracted using the correlation analysis were revealed effectively using the 178 CCM method (the  $\rho$  value between two variables was 0). Secondly, some weak coupling, 179 which was hardly detected using the correlation analysis (the correlation between the two variables were not significant), was extracted using the CCM method (a small  $\rho$  value). 180 181 Meanwhile, as Sugihara et al. (2012) suggested, the correlation between two variables 182 could be influenced significantly by other agent variables and thus the value of correlation 183 coefficient between two variables could not reflect the actual causality between them. 39

|-------------------------|

|---|---------------------|
|   |                     |
| Y |                     |

|---------------------------|---------------------------------|
|                           | 批注 [Office1]: Space after x(t), |
| () / Y                    | 带格式的: 字体: 倾斜             |
| Ύ                         |                                 |

Chen et al. (2017) further revealed that the correlation coefficient between individual meteorological factors and PM2.5 concentrations was usually much larger than the  $\rho$ value. This indicated that the causality of individual meteorological factors on PM2.5 concentrations was generally overestimated using the correlation analysis, due to the influences from other meteorological factors. In this case, the CCM method is an appropriate tool for quantifying bidirectional interactions between PM2.5 concentrations and individual meteorological factors in complicated atmospheric environment.

**191 4 Results**

192 Seasonal variations of PM2.5 concentrations have been proved by a large body of studies 193 (Cao et al., 2012; Shen et al., 2014; Yang and Christakos, 2015; Wang et al., 2015; Chen 194 et al., 2015; Chen, Y. et al. 2016; Chen, Z. et al., 2016). Hence, the research period was 195 divided into four seasons. According to traditional season division for China, spring was 196 set as the period between March 1st, 2014 and May 31st, 2014; summer was set as the 197 period between June 1st, 2014 and August 31st, 2014; autumn was set as the period between 198 September 1st, 2014 and November 30th, 2014; and winter was set as the period between 199 December 1st, 2014 and February 28th, 2015. For each city, the bidirectional coupling 200 between individual meteorological factors and PM2.5 concentrations in different seasons 201 was analyzed respectively using the CCM method. The CCM method is highly automatic 202 and only few parameters need to be set for running this algorithm: E (number of 203 dimensions for the attractor reconstruction),  $\tau$  (time lag) and b (number of nearest 204 neighbors to use for prediction). The value of E can be 2 or 3. A larger value of E produces 205 more accurate convergent maps. The variable b is decided by E(b = E + 1). A small value 206 of  $\tau$  leads to a fine-resolution convergent map, yet requires much more processing 207 time. Through experiments, we found that the final results were not sensitive to the 208 selection of parameters and different parameters mainly exerted influences on the 209 presentation effects of CCM. In this research, to acquire optimal presentationinterpretation 210 effects of convergent cross maps, the value of  $\tau \tau$  was set as 2 days and the value of E 211 was set 3. For each meteorological factor, its causality coupling with PM2.5 concentrations 212 can be represented using a convergent map. Since it is not feasible to present all these 213 convergent maps here, we simply display some exemplary maps to demonstrate how CCM 214 works (Fig 1).

---

## Referee Report (RR1)

**General Comments**

The authors have made a good effort to respond to the reviewer comments. My main remaining concern is that I still think that the paper as written is mostly descriptive and I think it would benefit from more in depth discussion of the significance of the work. I've pointed out a few specific areas where I think discussion could be added/improved in the comments below.

**Specific Comments**
pp 2, ln 38-42 – I wonder if it would be more effective to frame the opening here in terms of the air pollution itself, rather than haze, which I think of as being one of the side effects of air pollution.
pp 2, ln 47 – Similarly, I think the opening would be stronger if you provided more specifics about the health impacts of PM
pp 3, ln 68 – Is this ozone and PM or just PM?
pp 15, ln 352-354 – It would be nice to have some discussion in the paper (not necessarily in this section) of why different factors are more important in different regions
pp 15, ln 365-369 – Similarly, I think it would be valuable here to have some discussion of why winds are important in the Beijing-Tianjin-Hebei, but less important in summer. I'd imagine it has to do with having stronger, and more large-scale, circulation in winter?
pp 20, ln 416-420 – Why do you think precip might be more important in places with lower pollution levels? Is the air cleaner because the regions are wetter, or for another reason?
pp 22, footnote 6- For completeness, I think you should probably report the correlation coefficients or cite a paper that does.
pp 24, ln 548 – 552 – I'm still not convinced of this point. I think there is a difference between showing which meteorological factor is most dominant vs. showing sensitivity to changes in meteorological factors (for example, the anthropogenic greenhouse effect is dominated by CO2, but the atmosphere is much more sensitivity to changes in CH4). You said in your response that you looked at year-to-year variability, but I didn't see any discussion of it in the paper. Did it add any insights?
pp 24-25, conclusions section – I think this section could be deepened by making clear how this study fits in with the existing literature. How do these results compare to the literature, and what new knowledge has been added? As someone who is admittedly not well versed in the literature on air quality-meteorology interactions in China, this was not clear to me.

**Technical corrections**
pp 1, ln 20-22 – The wording of this sentence is unclear.
pp 2, ln 63 – should read "ozone concentrations were linearly correlated…"
pp 3, ln 66 – should read "… during the summer monsoon."
pp 3, ln 86 – should read " … humidity and solar radiation…"
pp 5, ln 141 – should read "8am-8pm"
pp 9 – Figure 1 appears blurry in the pdf file.

pp 13, ln 312-313 – should read "… are influenced by similar dominant meteorological factors…"

pp 14, ln 318 – should read "the higher the local PM2.5 concentrations,"

pp 14, ln 322 – should read "spring and summer are comparatively low."

pp 20, ln 410-413 – this sentence is a bit unclear.

pp 22, ln 501 – this wording is unclear.

---

## Author Response (AR2)

**Dear Dr Sally E. Pusede:**

**Thanks so much for giving us a chance to revise and submit our manuscript for**
**the potential publication in ACP. According to the reviewer's comment, we realized**
**that more in-depth analysis, especially those studies related to this research and**
**well explained some phenomena proposed in this research should be added. In the**
**revised manuscript, we have added a large body of references and much more**
**discussion and explanation to the revised manuscript according to the comments.**
**All the descriptive discussion in the previous manuscript, has been replaced with**
**more in-depth discussion, as well as the comparison with previous studies. By**
**explaining these previous studies, readers can see that the findings from this**
**research is consistent with and a major extension of previous studies. Meanwhile,**
**for some specific issues raised by the reviewer (e.g. the reason for variations of**
**meteorological influences across China, and the sensitivity of $PM_{2.5}$ variations to**
**the change of different meteorological factors, e.g. wind speed and precipitation),**
**we have added some references to explain and prove our suggestions with previous**
**case studies and experiments. By making these revisions, we believe this**
**manuscript has been significantly improved. Thanks again for the valuable**
**comments from you and the reviewer. We are more than willing to further revise**
**this manuscript if additional comments are given.**

**With regards**
**Ziyue**

General Comments

The authors have made a good effort to respond to the reviewer comments. My main
remaining concern is that I still think that the paper as written is mostly descriptive and I
think it would benefit from more in depth discussion of the significance of the work. I've
pointed out a few specific areas where I think discussion could be added/improved in the
comments below.

**R: Dear Reviewer, thanks so much for your valuable suggestions. By revising this manuscript according to your general comments, we believe we have replaced all those descriptive discussion with in-depth discussion, supported by previous studies and the manuscript has been greatly improved. Thanks again for your time and help. We are more than willing to further revise this manuscript if additional comments are given.**

Specific Comments pp 2, ln 38-42 – I wonder if it would be more effective to frame the opening here in terms of the air pollution itself, rather than haze, which I think of as being one of the side effects of air pollution.

**R: Thanks so much for this comment. In the revised manuscript, we have replaced the word "haze" with "high $PM_{2.5}$ concentrations" or "$PM_{2.5}$ pollution".**

pp 2, ln 47 – Similarly, I think the opening would be stronger if you provided more specifics about the health impacts of PM

**R: This is a very good suggestion. In the revised manuscript, we have added some details concerning specific influence of $PM_{2.5}$ concentration on human health as follows:**

Garrett and Casimiro, (2011) revealed that the relative risk for cardiovascular disease-related mortality for alder groups (>65 years) was 2.39% (95%C.I. 1.29%, 3.50%) for each 10 μg/m$^3$ $PM_{2.5}$ increase. Guaita et al. (2011) Qiao et al. (2014) found an interquartile range increment in $PM_{2.5}$ concentration (36.47 μg/m$^3$) led to a 0.57% [95% confidence interval (CI): 0.13%, 1.01%] increase in emergency room visits. Through experiments in nine French cities, Pasca et al. (2014) observed a notable effect of $PM_{2.5}$ (+0.7%, [−0.1; 1.6]) on all year non-accidental mortality for all age groups. In five European cities, estimation results suggested that a 12.4 μg/m$^3$ increase in the $PM_{2.5}$ concentration can lead to 3.0% [− 2.7%; 9.1%] increase in cardiovascular mortality (Lanzinger et al.,

**2015). Li et al. (2015) found that temperature played an important role in PM$_{2.5}$**
**induced mortality in Beijing. Under the condition of the lowest temperature**
**range (-9.7~2.6 °C), a 10 μg/m$^3$ increase in PM2.5 concentration led to an**
**increase of 1.27 % (95 % CI 0.38~2.17 %) in the relative risk (RR) of**
**cardiovascular mortality, which was the highest for all temperature ranges.**
pp 3, ln 68 – Is this ozone and PM or just PM?
**R: We are sorry that we did not make this part clear. This should PM, ozone and NO2**
**and we have revised this sentence in the revised manuscript.**
pp 15, ln 352-354 – It would be nice to have some discussion in the paper (not necessarily in
this section) of why different factors are more important in different regions.
**R: Thanks so much for this valuable comment. Yes, we should add more discussion,**
**especially the comparison with existing literatures to better explain this issue. We**
**have added a large body of relevant references and some possible explanations to**
**discuss the reasons why different factors are more important in different regions.**
**The explanation added to the revised manuscript: Firstly, the meteorological**
**conditions varied significantly in different regions across China. Secondly,**
**interactions between meteorological factors and PM$_{2.5}$ concentrations can be**
**highly complicated, subject to meteorological conditions and local PM$_{2.5}$**
**concentrations. A large body of references and relevant explanation included in the**
**revised manuscript is listed as follows, marked red:**
**The finding from this research was consistent with and a major extension of that**
**from previous studies by quantifying the influence of individual meteorological**
**factors in a large number of cities across China, instead of several scattered cities,**
**using a more robust causality analysis method, other than the correlation**
**analysis. Similar to previous studies, this study also revealed notable differences**
**in meteorological influences on PM$_{2.5}$ concentrations at the national scale, the**
**major reason for which was different meteorological conditions and complicated**

mechanisms of PM2.5-meteorology interactions. Firstly, notable differences existed in meteorological conditions across China. For instance, in winter, the frequency and intensity of precipitation are much higher and stronger in coastal areas than those in the North China region, where the frequency of strong winds is high in winter. Therefore, precipitation exerts a large influence on $PM_{2.5}$ concentrations in coastal regions whilst wind is the key influencing factor for $PM_{2.5}$ concentrations in the North China region in winter. Secondly, in addition to the large variations in the values of correlation coefficients, the interaction mechanisms between individual meteorological factors and $PM_{2.5}$ concentrations may also vary significantly across regions. For such meteorological influences as wind speed, its negative effect on $PM_{2.5}$ concentrations was consistent in China (He e al., 2017). On the other hand, He et al. (2017) suggested that temperature and humidity were either positively or negatively correlated with $PM_{2.5}$ concentrations in different regions of China. In terms of humidity, when the humidity is low, $PM_{2.5}$ concentration increases with the increase of humidity due to hygroscopic increase and accumulation of $PM_{2.5}$ (Fu et al., 2016). When the humidity continues to grow, the particles grow too heavy to stay in the air, leading to dry (particles drop to the ground) (Wang, J., & Ogawa, S. (2015)) and wet deposition (precipitation) (Li et al., 2015b), and the reduction of $PM_{2.5}$ concentrations. Similarly, there may be thresholds for the negative influences of precipitations on $PM_{2.5}$ concentrations (Luo et al., 2017). Heavy precipitation can have a strong washing-off effects on $PM_{2.5}$ concentrations and notably reduce PM2.5 concentrations. Meanwhile, slight precipitation may not effectively remove the high-concentration PM2.5. Instead, the slight precipitation may induce enhanced relative humidity and thus lead to the increase of $PM_{2.5}$ concentrations. Meanwhile, the washing-off effect from the same amount of precipitation on $PM_{2.5}$ concentrations in Xi'an, a city with higher $PM_{2.5}$ concentrations, was lower than that in Guangzhou (Guo et al., 2016), indicating local $PM_{2.5}$ concentrations also exerted a key role in the negative effects of precipitation. Meanwhile,temperature can either be negatively correlated with $PM_{2.5}$ concentrations by accelerating the flow circulation and promoting the dispersion of $PM_{2.5}$ (Li et al., 2015b), or positively correlated with $PM_{2.5}$ concentrations through inversion events (Jian et al., 2012).

pp 15, ln 365-369 – Similarly, I think it would be valuable here to have some discussion of why winds are important in the Beijing-Tianjin-Hebei, but less important in summer. I'd imagine it has to do with having stronger, and more large-scale, circulation in winter?

**R:Thanks so much for this suggestions. Yes, you are right. Large –scale circulation**

**in winter is the major reason for strong winds in winter. We added to two**

**references to the revised manuscript to prove this.**

**The results show that the presence of high air pressure in northwest Beijing**

**(NW-High) generally produced strong northwest winds with clean upwind air. As a**

**result, the NW-High played an important role in cleaning Beijing's PM$_{2.5}$ .(Tie et al.,**

**2015). In spring and winter, with strong northwesterly synoptic winds, the**

**sea-breeze circulation is confined in the coastal area, and the MPC is suppressed.**

**(Miao et.al, 2015)**

**Miao, Y., X.-M. Hu, S. Liu, T. Qian, M. Xue, Y. Zheng, and S. Wang (2015), Seasonal**

**variation   of local atmospheric circulations and boundary layer structure in the**

**Beijing-Tianjin-Hebei**

**region and implications for air quality, J. Adv. Model. Earth Syst., 7, 1602–1626,**

**Tie, X., Zhang, Q., He, H., Cao, J., Han, S., & Gao, Y., et al. (2015). A budget analysis**

**of the formation of haze in beijing. Atmospheric Environment, 100, 25-36.**

pp 20, ln 416-420 – Why do you think precip might be more important in places with lower pollution levels? Is the air cleaner because the regions are wetter, or for another reason?

**R: This is a very good point. You proposed a comment above that why**

**meteorological influences on PM$_{2.5}$ concentrations varies across China. And**

**precipitation is a good example that exerts different influences on PM$_{2.5}$**

**concentrations. We have added some relevant references to the revised**

**manuscript to address the question.**

**Luo et al. (2017). pointed out that there may be thresholds for the negative**
**influences of precipitations on PM$_{2.5}$ concentrations** Heavy precipitation can have
a strong washing-off effects on PM2.5 concentrations and notably reduce PM2.5
concentrations. Meanwhile, slight precipitation may not effectively remove the
high-concentration PM2.5. Instead, the slight precipitation may induce enhanced
relative humidity and thus lead to the increase of PM2.5 concentrations. So either
large amount of precipitation or low PM$_{2.5}$ concentrations can lead to a large
influence of precipitation on PM$_{2.5}$ concentrations. This was also proved by some
experiments. Through experiments, it was revealed that the washing-off effect
from the same amount of precipitation on PM2.5 concentrations in Xi'an, a city with
higher PM2.5 concentrations, was lower than that in Guangzhou, a city with lower
PM$_{2.5}$ concentrations. (Guo et al., 2016). In this research, we found that
precipitation mainly exerted a major role in influencing PM$_{2.5}$ concentrations in
coastal areas and Yangtze River Basins, which is consistent with these findings.
The precipitation in coastal areas and Yangtze River Basins is much larger than that
in North parts of China, where the PM$_{2.5}$ concentrations are higher and thus the
washing-off effects of precipitations are significantly limited. Studies in Nanjing
(Chen, T. et al., 2016), a mega city in the Yangtze River, and Hong Kong (Fung et
al.,), a mega coastal city, also proved that precipitation is the most important
meteorological factors for PM$_{2.5}$ concentrations.
**In the revised manuscript, the following text has been added:**
Taking the precipitation as an example. Luo et al. (2017). pointed out that there
may be thresholds for the negative influences of precipitations on PM2.5
concentrations and Guo et al. (2016) found that the same amount of precipitation
led to a washing-off effects in areas with higher PM2.5 concentrations. Hence,
precipitation mainly exerts a dominant influence on local PM2.5 concentrations
in winter for Yangtze River Basin or coastal cities, where the amount of
precipitation is large and the PM2.5 concentration is low, whilst precipitation
exerts a limited role in northern China, where the amount of precipitation is
small and the PM2.5 concentration is high.

pp 22, footnote 6- For completeness, I think you should probably report the correlation
coefficients or cite a paper that does.
**R: Thanks so much for this comment. In the revised manuscript, a reference, Chen**
**et al. (2017), which introduced some calculated correlation coefficients, as well as**
**the p value, has been cited.**
pp 24, ln 548 – 552 – I'm still not convinced of this point. I think there is a difference between
showing which meteorological factor is most dominant vs. showing sensitivity to changes in
meteorological factors (for example, the anthropogenic greenhouse effect is dominated by
CO2, but the atmosphere is much more sensitivity to changes in CH4). You said in your
response that you looked at year-to-year variability, but I didn't see any discussion of it in the
paper. Did it add any insights?
**R: Thanks so much for this point. With this comment, as well as some other**
**comments, we realized that we did not make this part clear. And the CCM method**
**itself may not well reveal the sensitivity of PM$_{2.5}$ variations to the change of**
**meteorological factors. So in the revised manuscript, firstly, we added some**
**relevant studies, which employed local scale analysis to prove a similar finding**
**from this research (e.g. the major influencing factor for Beijing in winter is wind**
**speed whilst the factor is precipitation in Yangtze River Basin and coast areas, such**
**as Nanjing, Guangzhou). In addition, some other issues, such as why strong wind is**
**prevailing in Beijing is in winter and why precipitation has a large washing-off**
**effects in coastal areas, has been added as well in above responses. Most**
**importantly, we added several papers that well proved the hypothesis presented**
**here. In terms of wind speed, in North China, PM2.5 concentration is much more**
**sensitive to the change of wind speed than that of other meteorological factors**
**(Gao et al., 2016). Meanwhile, wind-speed induced climate change led to the**
**change of PM2.5 concentrations by as much as 12.0 µgm-3, compared with the**
**change of PM2.5 concentrations by up to 4.0 µgm-3 in south-eastern, northwestern**
**and south-western China (Tai et al., 2010). Therefore, wind speed is BOTH the**
**dominant influencing factor and the factor that PM$_{2.5}$ variations are most sensitive**
**to in North China, and thus meteorological means for encouraging strong winds are**
**more likely to reduce PM2.5 concentrations considerably in North China. Similarly,**
**Luo et al. (2017) revealed that there was a threshold for precipitation to have**

washing-off effects for $PM_{2.5}$ concentrations and Guo et al. (2016) revealed that the same amount of precipitation had a stronger washing-off effect in areas with lower

$PM_{2.5}$ concentrations. Therefore, meteorological means for inducing precipitation are more likely to improve air quality in coastal cities and cities within the Yangtze

River basin, where there is a large amount of precipitation and relatively low PM2.5

concentrations.

In the revised manuscript, the following text has been added to the discussion and conclusion part:

For the heavily polluted North China region, especially the

Beijing-Tianjin-Hebei region, the northwesterly synoptic wind(Tie et al., 2015;

Miao et al., 2015)is much stronger in winter than winds in summer and exerts a dominant influence on $PM_{2.5}$ concentrations (Chen et al., 2017). Furthermore, in

North China, $PM_{2.5}$ concentration is much more sensitive to the change of wind speed than that of other meteorological factors (Gao et al., 2016). Meanwhile, wind-speed induced climate change led to the change of $PM_{2.5}$ concentrations by as much as 12.0 $\mu gm^{-3}$, compared with the change of $PM_{2.5}$ concentrations by up to 4.0 $\mu gm^{-3}$ in south-eastern, northwestern and south-western China (Tai et al.,

2010). Considering the strong winds in winter, the dominant influence of wind speed on $PM_{2.5}$ concentrations and the sensitivity of $PM_{2.5}$ feedbacks to the change of wind speed, meteorological means for encouraging strong winds are more likely to reduce $PM_{2.5}$ concentrations considerably in North China.

Similarly, Luo et al. (2017) suggested that only precipitation with a certain magnitude can lead to the washing-off effect of $PM_{2.5}$ concentrations whilst Guo et al. (2016) revealed that the variation of $PM_{2.5}$ concentrations was more sensitive to the same amount of precipitation in areas with lower $PM_{2.5}$

concentrations. Therefore, meteorological means for inducing precipitation are more likely to improve air quality in coastal cities and cities within the Yangtze

River basin, where there is a large amount of precipitation and relatively low

$PM_{2.5}$ concentrations

pp 24-25, conclusions section – I think this section could be deepened by making clear how this study fits in with the existing literature. How do these results compare to the literature, and what new knowledge has been added? As someone who is admittedly not well versed in the literature on air quality-meteorology interactions in China, this was not clear to me.

**R: Thanks so much for this valuable comment. As explained above, in the discussion**

**part, we have added a large part of literature review, concerning existing research**

**on the correlations between individual meteorological factors and PM$_{2.5}$**

**concentrations. As you can see, previous studies mainly focused on the**

**meteorological influences on PM$_{2.5}$ concentrations in several specific cities, and**

**thus a comprehensive comparison has yet been conducted. Furthermore, previous**

**studies mainly employed the correlation analysis, which may be biased in**

**quantifying meteorological influences on PM$_{2.5}$ concentrations. The findings from**

**this research was generally consistent with that from previous studies Firstly, by**

**comparing the results from different studies, one can see that meteorological**

**influences varied significantly in different cities across China, which was further**

**proved by this research conducted at the national scale. Secondly, the extracted**

**dominant meteorological factors for some cities in most polluted winter (e.g. wind**

**for Beijing, precipitation for Nanjing and Hong Kong) were similar to findings from**

**this research. This research is a major extension of previous studies by extending**

**the study sites from scattered cities to 188 cities all over China. In this case,**

**regional similarity in meteorological influences on PM$_{2.5}$ concentrations, which**

**cannot be extracted based on local scale case studies, was revealed. Meanwhile,**

**the CCM method provides more reliable quantitative analysis of meteorological**

**influences on PM$_{2.5}$ concentrations, compared with the correlation coefficient.**

**Thirdly, this research analyzed a complete set of eight meteorological factors,**

**whilst previous studies generally focused on a smaller number of meteorological**

**factors. Fourthly, the study period of previous studies conducted in specific cities**

**varied significantly, ranging from weeks to years. Instead, this research used a**

**unified three-year observation data to compare meteorological influences on PM$_{2.5}$**

**concentrations in different cities, and thus the comparison result is more robust.**

**Finally, some statistics of the general influence of individual meteorological factors**

**(e.g. wind, precipitation) was rarely compared and thus the three major factors,**
**temperature, humidity and wind for PM$_{2.5}$ concentrations all over China, revealed in**
**this research, were a major contribution to the existing literatures.**
**So compared with previous studies, this research further proved the diversity of**
**meteorological influences on PM$_{2.5}$ concentrations in China, and revealed some**
**regional patterns which were rarely studied, and provided decision makers with a**
**comprehensive understanding of meteorological influences across China, which is**
**of practical significance for management of local and regional air quality.**
**Thanks again for this valuable comment. By responding to this comment, we**
**reviewed many relevant papers, and better linked this research to other studies.**
**This manuscript, especially the discussion and conclusion part, has been improved**
**significantly.**
**In the revised manuscript, the following text has been added to the discussion and**
**conclusion part:**
**Discussion:**
Despite the lack of a comprehensive comparison of meteorological influences on
PM$_{2.5}$ concentrations in different regions, some studies concerning
meteorology-PM$_{2.5}$ relationship in specific areas have been conducted and
correlations between individual meteorological factors and PM$_{2.5}$ concentrations
have been analyzed in such mega cities as Nanjing ( Chen, T. et al., 2016; Shen and
Li., 2016;), Beijing (Huang et al., 2015; Yin et al., 2016), Wuhan ( Zhang et al.,
2017), Hangzhou (Jian et al., 2012), Chengdu ( Zeng and Zhang, et al. 2017) and
Hong Kong (Fung et al., 2014). These studies mainly employed correlation
analysis to quantify the influence of several meteorological factors on PM$_{2.5}$
concentrations and suggested that meteorological influences on PM$_{2.5}$
concentrations varied significantly across regions. The dominant meteorological
factors for P$_{2.5}$ concentrations (presented as the largest correlation coefficients in
previous studies and the $\rho$ value in this research ) demonstrated notable regional differences. For Nanjing (Chen, T. et al., 2016), a mega city in the Yangtze River,
and Hong Kong (Fung et al.,), a mega coastal city, precipitation exerted the
strongest influence whilst wind speed exerted a weak influence on $PM_{2.5}$
concentrations in winter. On the other hand, for winter, wind speed was the
dominant meteorological factor for $PM_{2.5}$ concentrations in Beijing (Huang et al.,
2015.) , a mega city in North China, and precipitation played a weak role in
affecting local $PM_{2.5}$ concentrations . These studies generally analyzed and compared
the influences of different meteorological factors on $PM_{2.5}$ concentrations and
extracted the dominant meteorological influencing factors for specific areas. However,
most studies were conducted at the local scale and few studies have focused on the
comparison and statistics of meteorological influences on $PM_{2.5}$ concentrations in
different areas. Meanwhile, although the correlation coefficient can be used to
understand and compare the general magnitude of the influence of individual
meteorological factors, the correlation analysis, as explained above, may lead to large
bias in quantifying the meteorological influences on $PM_{2.5}$ concentrations.

**Conclusion part:**
Previous studies examined the correlation between individual meteorological
influences and PM2.5 concentrations in some specific cities and the comparison
between these studies indicated that meteorological influences on PM2.5
concentrations varied significantly across cities and seasons. However, these
scattered studies conducted at the local scale cannot reveal regional patterns of
meteorological influences on PM2.5 concentrations. Furthermore, previous
studies generally selected different research periods and meteorological factors,
making the comparison of findings from different studies less robust. Thirdly,
these studies employed the correlation analysis, which may be biased significantly
due to the complicated interactions between individual meteorological factors.
This research is a major extension of previous studies.  Based on a robust
causality analysis method CCM, we quantified and compared the influence of
eight meteorological factors on local PM2.5 concentrations for 188 monitoring
cities across China using PM2.5 and meteorological observation data from 2014.3
to 2017.2. Similar to previous studies conducted at the local scale, this research
further proved that meteorological influences on PM2.5 concentrations were of notable seasonal and spatial variations at the national scale. Furthermore, this
research revealed some regional patterns and comprehensive statistics of the
influence of individual meteorological factors on PM2.5 concentrations, which
cannot be understood through small-scale case studies.
Technical corrections
pp 1, ln 20-22 – The wording of this sentence is unclear.
**R:** This sentence has been revised from "For the heavily polluted North China region, the
higher PM2.5 concentrations, the larger influences meteorological factors exert on PM2.5
concentrations."
To "For the heavily polluted North China region, when PM2.5 concentrations are high,
meteorological influences on PM2.5 concentrations are strong."
pp 2, ln 63 – should read "ozone concentrations were linearly correlated…"
**R: Corrected.**
pp 3, ln 66 – should read "… during the summer monsoon."
R: Corrected.
pp 3, ln 86 – should read " … humidity and solar radiation…"
**R: Corrected.**
pp 5, ln 141 – should read "8am-8pm"
**R: Corrected.**
pp 9 – Figure 1 appears blurry in the pdf file.
**R: This Figure 1 has been reproduced.**
pp 13, ln 312-313 – should read "… are influenced by similar dominant meteorological
factors…"
**R: Corrected.**
pp 14, ln 318 – should read "the higher the local PM2.5 concentrations,"
**R: Corrected.**
pp 14, ln 322 – should read "spring and summer are comparatively low."
**R: Corrected.**
pp 20, ln 410-413 – this sentence is a bit unclear.
**R: Thanks so much for this. We changed the sentence from,** "As a result, when
analyzing meteorological influences on local $PM_{2.5}$ concentrations for a specific city,
the influence of meteorological factors that have little influence on $PM_{2.5}$

concentrations at a large scale should be carefully examined at the local scale. "

to "As a result, when analyzing meteorological influences on local $PM_{2.5}$

concentrations for a specific city, meteorological factors that have little influence on

$PM_{2.5}$ concentrations at a large scale should also be comprehensively considered"

pp 22, ln 501 – this wording is unclear.

R: in the revised manuscript, we changed the sentence from "dynamic statistical models comprehensively consider the meteorological influences on $PM_{2.5}$ concentrations"

[revised manuscript text omitted]

---

## Author Response (AR3)

To Dr Sally Pusede:

**R: Thanks so much for providing us some many valuable comments on our manuscript. Through two rounds of major revision, many new content has been added to the manuscript, and thus the focus of the manuscript has been changed significantly. Hence, the introduction and discussion part should be revised accordingly. As a result, your valuable comments on the structural revision on the introduction and discussion part are so important, and we do realize that your suggestion help to improve this manuscript significantly. In addition to these structural revision suggestions, we also fully revised this manuscript according to all your general and detailed comments, including figure revision, and details on the methodology and data explanation. Furthermore, more quantitative discussion has also been added to corresponding parts, according to your suggestions. Thanks again for processing and carefully reviewing this manuscript. We are more than willing to conduct additional revisions if you have further revision suggestions.**

The manuscript is improved, but please address the concerns below prior to publication.

The Introduction does not properly frame the analysis. As I understand it, the manuscript describes an application of CCM to regional-scale relationships between PM2.5 and meteorological variables and discusses these results in the context of previous statistical approaches. There is no need for elementary detail on PM2.5 health effects or on trends in pollutants other than PM2.5 (e.g., ozone), especially outside of China. Focus the introductory content on material relevant to the manuscript, especially research on PM2.5-meteorology relationships in China. The CCM method should be introduced, since application of CCM is the heart of the work. The paragraph summarizing past work on PM2.5-meteorology correlations needs to be edited (begins line 104). The first two sentences are unnecessary, replace them with a sentence summarizing what has been observed so the reader is not presented with a listing of past results without context. I

encourage you to add a concluding paragraph to the Introduction that describes the analysis that follows.

**R: Thanks so much for your detailed suggestions on revising the introduction part. Yes, the introduction part should focus more on the previous studies concerning meteorology-PM$_{2.5}$ concentrations in China, and thus we have significantly reduced the introduction of PM$_{2.5}$ induced diseases and the meteorological influences on PM$_{2.5}$ concentrations in other countries. Furthermore, the interactions between other airborne pollutants (e.g. O$_3$ and PM$_{10}$) and meteorological factors have been completely deleted in the revised manuscript. Meanwhile, more relevant studies concerning meteorological influences on PM$_{2.5}$ concentrations in China have been added to the revised manuscript. These studies examined the correlation between PM$_{2.5}$ concentrations and different meteorological factors in specific cities. However, findings from these studies conducted at a local scale cannot reveal regional and national patterns of meteorological influences on PM2.5 concentrations in China. In addition, these studies mainly employed short-term observation data (e.g. one season or one year) and thus revealed characteristics of meteorological influences on PM$_{2.5}$ concentrations may be biased by inter-annual variations.**

**The correlation analysis employed in previous studies may lead to mirage correlations and can be biased significantly by influences from other variables. So it is necessary to introduce the CCM method briefly. A short explanation of CCM, especially its advantages compared with the correlation analysis, was added to the introduction. Finally, according to your comment, we added a conclusion part to the introduction part that describes the following analysis.**

**The added text in the revised manuscript is as follows:**

[revised manuscript text omitted]

I am not familiar with meteorological measurement evaporation. Please clarify. The footnote is not helpful and is not encouraged in ACP.

**R: Thanks so much for this comment. The evaporation measurement has been added to the revised manuscript and the use of footnote has been removed.**

**The added explanation of evaporation is as follows:**

Evaporation indicates the amount of evaporation-induced water loss during a certain period and is usually calculated using the depth of evaporated water in a container. For this research, small (large) evaporation indicates the amount of evaporated water measured using a container with a diameter of 10cm (30cm) during 24 hours (unit: mm). Generally, the measured values using the two types of equipment are of slight differences.

The meteorological variables should not be italicized.

**R: Corrected**

Line 157: Write out the word "minimum."

**R: Corrected**

Variable abbreviations like meanTEM and maxPRS reduce readability, rather than improve it. I recommend they are all removed.

**R:Thanks so much for this comment. We have removed all these inappropriate abbreviations.**

Page 6: Footnotes should be avoided. Place the information in the main text.

**R: All these footnotes have been removed from the manuscript and placed in the main text.**

Line 175: Remove etc., instead begin the list with "e.g."

**R:All etc. in the manuscript have been replaced with "e.g.".**

Line 234: Say where.

R: In the revised manuscript, we have specified these locations with seasonal variations of $PM_{2.5}$ concentrations.

Seasonal variations of $PM_{2.5}$ concentrations have been revealed in Beijing (Chen et al., 2015; Chen, Y. et al., 2016; Chen, Z. et al., 2016), Nanjing (Shen et al., 2014), Shandong Province (Yang and Christakos, 2015) and the Beijing-Tianjin-Hebei region (Wang et al. 2015; Chen, Z. et al., 2017). In addition to these local and regional studies, Cao et al. (2012) further compared seasonal variations of $PM_{2.5}$ concentrations in seven southern cities (Chongqing, Guangzhou, Hong Kong, Hangzhou, Shanghai, Wuhan, and Xiamen) and seven northern cities (Beijing, Changchun, Jinchang, Qingdao, Tianjin, Xi'an, and Yulin) across China.

Avoid use of the word "prove."

**R: All use of the word "prove" has been removed.**

Remove all uses of the word "haze." Be specific, if you mean PM2.5, say that.

**R: All the use of "haze" has been removed in the revised manuscript.**

The first sentence of many paragraphs in the paper is superfluous and should be deleted to improve readability.

**R: Thanks again for this comment. We again reviewed this manuscript and deleted the first sentence of many paragraphs.**

Fig. 1: Use full titles and full axis labels so the figure is more easily read.

**R:This figure has been reproduced according to your comments.**

Fig. 1: Explain why these four panels were selected.

**R: We selected the winter 2014 in Beijing as an instance to demonstrate how the CCM figure explain the bi-directional coupling between meteorological factors and PM$_{2.5}$ concentrations. For winter, 2014, Beijing was one of heavily polluted cities with extremely high PM$_{2.5}$ concentrations and the calculated p value of meteorological factors on PM$_{2.5}$ concentrations was very large. So the coupling between PM$_{2.5}$ concentrations and meteorological factors in winter, 2014 is an ideal example to demonstrate how CCM works. To better present the effects of CCM method, we specifically selected four meteorological factors, which had the strongest influences on local PM$_{2.5}$ concentrations. Meanwhile, PM$_{2.5}$ concentrations also had notable feedback effects on these meteorological factors. By selecting these four major meteorological factors, the output CCM is more likely to provide readers a general comparison of the magnitude of simultaneous influences of meteorological factors on the local PM$_{2.5}$ concentration and its feedback effects. If other factors that exerted weak influences on PM$_{2.5}$ concentrations were selected, small p values would make the curves from exemplary CCMs difficult to understand and compare.**

**In the revised manuscript, we have added the following explanation:**

As a heavily polluted city, we presented the interactions between PM$_{2.5}$ concentrations and meteorological factors in Beijing in winter, when the local PM$_{2.5}$ concentration was the highest, as an example. Four major meteorological factors, wind, humidity, radiation and temperature, which exerted much stronger influences on PM$_{2.5}$ concentrations than other factors, were employed. Due to the strong bidirectional coupling between PM$_{2.5}$ concentrations and these meteorological factors, Figure 1 not only demonstrates how CCM output could be interpreted, but also provides readers with a general comparison of the magnitude of simultaneous influences of different meteorological factors on the local PM$_{2.5}$ concentration and its feedback effects.

Line 300: This entire paragraph can be deleted.

**R: We have deleted this paragraph.**

Figs 2 will not reproduce well. It is difficult to distinguish the gray scale. The gray scale limits should be rounded to integers. I recommend the wind roses be made larger and the legends labeled with words rather than abbreviations.

**R:Thanks so much for this comment. We have rounded the gray scale to integers and reproduced the legends with words as you suggested. We also tried to make these wind roses a bit larger, yet we cannot make these wind roses much larger. The reason is that we used a unified wind rose legend scale for all seasons to give readers a comparable presentation of how the magnitude of meteorological influences varied across different seasons and regions. Since the p value of different factors ranged from around 0.1 to 0.8 in different seasons, the size of rose pedals also varied significantly. I understand the size of wind roses in some regions in summer (or other seasons) is a bit small. However, if we further extend the legend scale of the wind roses, although those small wind roses can be presented better, there will be severe overlapping effects for those large wind roses for those representative cities in the North China plain in winter. Since a clear presentation of these large wind roses in heavily polluted cities is of great importance, some very small wind roses caused by extreme small p values cannot be further made bigger.**

Figs 2 and 3: Remove the map inset and the yellow dashed boundary. These do not contribute the display of the data.

**R: Thanks so much for this suggestion. We have removed the inset map and the yellow dashed boundary and this revision indeed improves the display of the data.**

Line 323: This paragraph is too vague. You are reporting on your quantitative analysis here. This text is too general and could be known without your CCM results.

**R: Thanks so much for this comment on this paragraph and other parts. We do realized that without quantitative p value presented here, the simple qualitative explanation is too vague. So in the revised manuscript, we included more details on the quantitative explanation of these patterns.**
**The following text has been added to the revised manuscript.**

Take several mega cities in different regions for instance. During 2014-2016, the three major meteorological influencing factors for PM$_{2.5}$ concentrations in Beijing, a mega city in the North China plain, were as follows: humidity (0.48),wind (0.37) and evaporation (0.31) for spring, humidity (0.39),temperature (0.34) and SSD (0.25) for summer, humidity (0.56),evaporation (0.51) and wind (0.41) for autumn, and humidity (0.76), wind (0.57) and evaporation (0.52) for winter. The three major meteorological influencing factors for PM$_{2.5}$ concentrations in Shanghai, a mega city in the Yangtze River Basin, were as follows: temperature (0.264), air pressure (0.260) and wind (0.25) for spring, temperature (0.40), wind (0.38) and humidity (0.27) for summer, temperature (0.39), wind (0.28) and humidity (0.17)  for autumn, and precipitation (0.36), wind direction (0.25) and humidity (0.19) for winter. The three major meteorological influencing factors for PM$_{2.5}$ concentrations in Wuhan, a major city in Central China region, were as follows: precipitation ( 0.18), wind (0.16) and temperature (0.09) for spring, humidity (0.47), temperature (0.41) and wind (0.34) for summer, wind (0.44), precipitation (0.31) and temperature (0.26) for autumn, and precipitation (0.33), temperature (0.19) and wind (0.15) for winter. The three major meteorological influencing factors for PM$_{2.5}$ concentrations in Guangzhou, a major city in Southern China region, were as follows: wind (0.31), precipitation (0.24) and air pressure (0.23) for spring, air pressure (0.51), temperature (0.41) and wind (0.37) for summer, temperature (0.47), wind (0.36) and precipitation (0.29) for autumn, and temperature (0.52), wind (0.48) and air pressure (0.33). Notable seasonal variations of meteorological influences on PM$_{2.5}$ concentrations were found in these mega cities across China.

Line 331: Same comment. This paragraph is too vague. You are reporting on your quantitative analysis here. This text is too general and could be known without your CCM results.

**R: Thanks so much for this comment. More detailed quantitative analysis result has been included in this part.**

**The following text has been added to the revised manuscript.**

Take four major cities, Beijing, Tianjin, Taiyuan and Shijiangzhuang, in the North China Plain for example. For winter, SSD, evaporation, humidity and wind were the major meteorological factors for PM$_{2.5}$ concentrations in the four cities and the $\rho$ value of these four factors was 0.50, 0.52, 0.76 and 0.57 for Beijing, 0.41, 0.44, 0.56

and 0.50 for Tianjin, 0.44, 0.36, 0.61 and 0.41 for Taiyuan, and 0.62, 0.58, 0.56 and 0.60 for Shijiazhuang respectively, presenting a similar regional pattern. Meanwhile, meteorological influences on PM$_{2.5}$ concentrations in cities within the Yangtze River Basin, especially the dominant factors, were also of some regional similarities. Take four major cities in the Yangtze River Basin, Shanghai, Nanjing, Hangzhou and Nanchang for example. For summer, precipitation, humidity, temperature and wind were the major meteorological factors for PM$_{2.5}$ concentrations in these four cities and the $\rho$ value of these factors was 0.21, 0.27, 0.40 and 0.38 for Shanghai, 0.29, 0.41, 0.34 and 0.33 for Nanjing, 0.28, 0.27, 0.23 and 0.27 for Hangzhou, and 0.24, 0.33, 0.21 and 0.29 for Nanchang. Despite of some differences in the $\rho$ values, similar dominant meteorological factors and the similar magnitude of meteorological influences demonstrated regional similarities of meteorological influences on PM$_{2.5}$ concentrations in the Yangtze River Basin.

Line 340: Same comment. This paragraph is too vague. You are reporting on your quantitative analysis here. This text is too general and could be known without your CCM results.

**R: Thanks so much for this comment. More detailed quantitative analysis result has been included in this part.**

**The following text has been added to the revised manuscript.**

Take four major cities in the North China region for instance. For Beijing, the major influencing meteorological factors in summer were temperature (0.34), humidity (0.39) and SSD (0.25) whilst the major influencing meteorological factors in winter were humidity (0.76), wind (0.57), evaporation (0.52) and SSD (0.5). For Tianjin, the major influencing meteorological factors in summer were precipitation (0.34), temperature (0.22) and air pressure (0.25) whilst the major influencing meteorological factors in winter  were humidity (0.76), wind (0.57), evaporation (0.52) and SSD (0.50). For Shijiazhuang, the major influencing meteorological factors in summer were SSD (0.4), humidity (0.26) and evaporation (0.26) whilst the major influencing meteorological factors in winter  were SSD (0.62), wind (0.60), evaporation (0.58) and humidity (0.56). For Taiyuan, the major influencing meteorological factors in summer were temperature (0.32), air pressure (0.23) and precipitation (0.20) whilst the major influencing meteorological factors in winter  were humidity (0.61), SSD

(0.44) and wind (0.41).

Line 352: This entire paragraph can be deleted.

**R:This paragraph has been fully removed in the revised manuscript.**

Line 366: Likewise, points a–c are quite general. Can you talk about these results in specific quantitative terms?

**R: Thanks so much for this comment. More detailed quantitative analysis result has been included in this part.**

**The following text has been added to the revised manuscript to a.**

Here we analyzed the $\rho$ value of precipitation in cities within the Yangtze River Basin and cities within the Beijing-Tianjin-Hebei region respectively. For winter, precipitation was the dominant factor for $PM_{2.5}$ concentrations in Shanghai, Hangzhou and Nanchang within the Yangtze River Basin and the $\rho$ value of precipitation was 0.36, 0.29 and 0.31 respectively. Meanwhile, the $\rho$ value of precipitation in Beijing, Tianjin and Shijiazhuang within the Beijing-Tianjin-Hebei region was 0.08, 0.01 and 0.06 respectively.

**The following text has been added to the revised manuscript to b.**

The prevalence of different meteorological factors across China can also be reflected according to the number of cities where this specific factor is the dominant factor for local $PM_{2.5}$ concentrations. For winter, the number of cities with temperature, wind or humidity as the dominant factor was 56,48 and 44 respectively. Meanwhile, the number of cities with SSD or wind direction as the dominant factor was 3 and 1 respectively.

**The following text has been added to the revised manuscript to c.**

Take some major cities in North China region for instance. For winter, the dominant meteorological factors for Beijing, Tianjin, Taiyuan, Zhangjiakou, Handan and Jining was humidity (0.76),humidity (0.56), humidity (0.61), wind (0.62), humidity (0.43) and humidity (0.52) respectively. Meanwhile, for summer, the dominant meteorological factors for Beijing, Tianjin, Taiyuan, Zhangjiakou, Baoding, Handan and Jining was humidity (0.39), precipitation (0.28), temperature (0.23), temperature (0.47), air pressure (0.21) and SSD (0.18).

Fig. 3: Same comments as on Fig 2. Can you make these figures more readable? Is the size of the symbol important? They appear to vary.

**R: Thanks so much for this comment. We have reproduced these symbols**

**and attempts to make different samples appear with a similar size.**

Line 458: Do you mean "across different regions?"

**R: Yes, and we have revised the use of "in" to " across".**

Lines 486–488: What is the evidence for this?

**R: Sorry that we did not make this clear in the previous manuscript. The main reason why local-scale studies cannot reveal regional patterns are as follows: a. Firstly, local-scale studies mainly focuses on specific cities and thus regional similarities of meteorological influences on $PM_{2.5}$ concentrations may not be revealed. E.g. a case study in Nanjing cannot reflect the spatio-temporal patterns in the Yangtze-River Basin. b. Previous local-scale studies were conducted at different time and thus findings from these studies could not be compared. c. Due to highly complicated interactions between meteorological factors in the atmospheric environment, the correlation coefficient between $PM_{2.5}$ concentrations and different meteorological factors was not a reliable indicator to compare across different regions. In this case, based on the CCM method, this research examined meteorological influences on $PM_{2.5}$ concentrations for 188 monitoring cities across China using a unified research period of three consecutive years and a unified set of meteorological factors and better understood the regional patterns of meteorological influences on $PM_{2.5}$ concentrations across China. The revised content was detailed explained in the following responses.**

Line 532: Much of the content of this paragraph appears to be irrelevant to the manuscript.

**R:This paragraph has been entirely deleted in the revised manuscript.**

The Discussion should be refocused. The purpose of this analysis was to apply CCM to a wide region. The Discussion should then consider the difference between the author's CCM results and past analyses on more local areas and to compare CCM to other statistical approaches. First, it is well known that PM2.5

correlates with meteorological variables. Second, ACP is not an appropriate journal to expound upon government policies and public response unrelated to analysis perform. These are not the discussion point, instead the authors must answer:

What new do we learn about PM2.5-meteological relationships by using CCM over a large spatial region?

**R: Thanks so much for this comment. We have fully revised the discussion part. The discussion on the government policies and public responses unrelated to analysis perform have been massively reduced or removed. We have added some new content concerning the comparison between this large-scale research using the CCM method. and previous local-regional scale research using the correlation analysis. The added content concerning "What new do we learn about PM2.5-meteological relationships by using CCM over a large spatial region" is as follows:**

[revised manuscript text omitted]

---

## Author Response (AR4)

To Dr Sally Pusede:

**Thanks so much for your careful checking of all our detailed revisions to the previous manuscript. We have again fully revised the manuscript according to your comments and conducted a thorough proofreading from authors and an editor. Thanks again for processing our manuscript and giving so many valuable comments.**

Line 286: write out meanRHU and maxWIN. Write out other instances as well.

**R: Sorry for these abbreviations. We have corrected them all in the revised manuscript.**

The organization with letter labeling (a., b., c.) should be changed to better adhere to ACP style conventions.

**R: Thanks so much for pointing this out. We have checked many newly published ACP papers and realized that the labeling (1,2,3) at the beginning is a more frequently used style in ACP. So we have adjusted the labeling in the revised manuscript.**

I appreciate the quantitative results added to the Results section. The section is now quite long, I recommend either removing rho values inessential to the main points, or adding a table to display the values to improve readability.

**R: Thanks so much for this. We also realized that some quantitative description is too long and prevented readers having a clear understanding of given information. Therefore, we have removed much of the quantitative text in the result part and added two tables to the revised manuscript as follows:**

Table 1 Major meteorological influencing factors for $PM_{2.5}$ concentrations in four mega cities within different regions

| City | Season | Three major meteorological factors | | |
|---|---|---|---|---|
| Beijing | Spring | Humidity(0.48) | Wind(0.37) | Evaporation(0.31) |
| | Summer | Humidity(0.39) | Temperature(0.34) | SSD (0.25) |
| | Autumn | Humidity(0.56) | Evaporation(0.51) | Wind (0.41) |
| | Winter | Humidity(0.76) | Wind(0.57) | Evaporation(0.52) |
| Shanghai | Spring | Temperature(0.264) | air pressure(0.260) | Wind (0.25) |
| | Summer | Temperature(0.40) | Wind(0.38) | Humidity(0.27) |
| | Autumn | Temperature(0.39) | Wind(0.28) | Humidity(0.17) |
| | Winter | Precipitation(0.36) | Wind direction(0.25) | Humidity(0.19) |
| Wuhan | Spring | Precipitation(0.18) | Wind (0.16) | Temperature(0.09) |
| | Summer | Humidity(0..47) | Temperature(0.41) | Wind (0.34) |
| | Autumn | Wind (0.44) | Precipitation(0.31) | Temperature(0.26) |
| | Winter | Precipitation(0.33) | Temperature(0.19) | Wind (0.15) |
| Guangzhou | Spring | Wind(0.31) | Precipitation(0.24) | Air pressure(0.23) |
| | Summer | Air pressure(0.51) | Temperature(0.41) | Wind (0.37) |
| | Autumn | Temperature(0.47) | Wind(0.36) | Precipitation(0.29) |
| | Winter | Temperature(0.52) | Wind(0.48) | Air pressure (0.33) |

Table 2 Summer and winter major influencing meteorological factors for $PM_{2.5}$ concentrations in four major cities in the North China region

| City | Season | Major influencing meteorological factors | | | |
|---|---|---|---|---|---|
| Beijing | Summer | humidity | temperature | SSD | |
| | | 0.39 | 0.34 | 0.25 | |
| | Winter | humidity | wind | evaporation | SSD |
| | | 0.76 | 0.57 | 0.52 | 0.50 |
| Tianjin | Summer | precipitation | air pressure | temperature | |
| | | 0.34 | 0.25 | 0.22 | |
| | Winter | humidity | wind | evaporation | SSD |
| | | 0.56 | 0.50 | 0.44 | 0.41 |
| Shijiazhuang | Summer | SSD | humidity | evaporation | |
| | | 0.4 | 0.26 | 0.26 | |
| | Winter | SSD | wind | evaporation | humidity |
| | | 0.62 | 0.60 | 0.58 | 0.56 |
| Taiyuan | Summer | temperature | air pressure | precipitation | |
| | | 0.32 | 0.23 | 0.20 | |
| | Winter | humidity | SSD | wind | |
| | | 0.61 | 0.44 | 0.41 | |

Is Fig. 4 necessary? it appears to provide the same in formation as Table 1.

**R: Yes, some important information (mean value, SD) can be seen from Table 3( Table 1 in the original manuscript). But the violin Chart (Fig 4) additionally provides some additional information of the influence of individual meteorological influence. The most important information from the Figure, which cannot be understood from the table, is the frequency of different p values amongst 188 cities (similar to, but better than a histogram). This important information can help readers have a better understanding of the distribution of the influence of meteorological factors on $PM_{2.5}$ concentrations across China. Therefore, we would appreciate that you could understand we prefer to keep this figure to give readers a comprehensive understanding of the statistical distribution of**

**meteorological influences on PM$_{2.5}$ concentrations across China.**

The manuscript would benefit from a thorough proofreading.

**R: A Thorough proofreading has been conducted by authors and expert editor.**

Line 539: PM2.5

**R: Corrected.**

I recommend the Discussion section be edited to be made slightly more concise, if possible.

**R: This is a very good point. After a careful check of the discussion part, we have deleted many redundant sentences and made the discussion more concise. Thanks so much for this valuable comment.**

[revised manuscript text omitted]